# Fairness in Social Influence Maximization via Optimal Transport

**Shubham Chowdhary**
ETH Zürich
schowdhary@ethz.ch

**Giulia De Pasquale**[*]
Eindhoven University of Technology
g.de.pasquale@tue.nl

**Nicolas Lanzetti**[*]
ETH Zürich
lnicolas@ethz.ch

**Ana-Andreea Stoica**
Max Planck Institute, Tübingen
ana-andreea.stoica@tuebingen.mpg.de

**Florian Dörfler**
ETH Zürich
dorfler@ethz.ch

## Abstract

We study fairness in social influence maximization, whereby one seeks to select seeds that spread a given information throughout a network, ensuring balanced outreach among different communities (e.g. demographic groups). In the literature, fairness is often quantified in terms of the expected outreach within individual communities. In this paper, we demonstrate that such fairness metrics can be misleading since they overlook the stochastic nature of information diffusion processes. When information diffusion occurs in a probabilistic manner, multiple outreach scenarios can occur. As such, outcomes such as "In 50% of the cases, no one in group 1 gets the information, while everyone in group 2 does, and in the other 50%, it is the opposite", which *always* results in largely unfair outcomes, are classified as fair by a variety of fairness metrics in the literature. We tackle this problem by designing a new fairness metric, *mutual fairness*, that captures variability in outreach through optimal transport theory. We propose a new seed-selection algorithm that optimizes both outreach and mutual fairness, and we show its efficacy on several real datasets. We find that our algorithm increases fairness with only a minor decrease (and at times, even an increase) in efficiency.

## 1 Introduction

**Problem Description.** Social networks play a fundamental role in the spread of information, as in the context of commercial products endorsement [17], job vacancy advertisements [3], public health awareness [27], etc. Information, ideas, or new products can either go viral and potentially bring significant changes in a community or die out quickly. In this context, a fundamental algorithmic problem arises, known as Social Influence Maximization (SIM) [11, 12]. SIM studies how to strategically select a pre-specified small proportion of nodes in the social network – the *early adopters* or *seeds* – so that the outreach generated by a diffusion process that starts at these early adopters is maximized. Consider, for example, a product endorsement campaign: the early adopters are strategically selected users who receive the product first to promote it to their friends, who in turn may or may not adopt it. The optimal selection of early adopters is known to be an NP-hard problem [11]. Thus, many heuristic strategies have been proposed, based on iterative processes such as greedy algorithms or on network centrality measures. However, all these algorithms purely rely on the graph topology and are agnostic to users' demographics, which raises significant fairness concerns, especially in contexts of health awareness campaigns, education, and job advertisements, where one wants to ensure an

---

[*]Authors contributed equally.

38th Conference on Neural Information Processing Systems (NeurIPS 2024).

equitable spreading of information. Indeed, real-world social networks are populated by different social groups based on gender, age, race, geography, etc., with different group sizes or connectivity patterns. Ignoring these aspects and focusing only on the outreach maximization process usually leads to the early adopters being the most central nodes. Consequently, low-interconnected minorities are often neglected from the diffusion process, thus causing fundamental inequity in the information propagation and biases exacerbation [10, 25].

**Related Work.** The problem of SIM was first introduced in 2003 in Kempe et al. [11], where the problem of optimally selecting a (limited) set of early adopters was proved to be NP-hard. The study of SIM under fairness guarantees has a more recent history [5]. Several multiple group-level fairness metrics have been proposed over the years [6]. They fall under the notions of *equity* [23, 9, 10], *equality* [6], *max-min fairness* [7, 30], *welfare* [16], and *diversity* [25]: all of them quantify the fair distribution of influence across groups. In particular, Stoica et al. [23] propose a new SIM algorithm that operates under the constraint that, in expectation, the same percentage of users in each category is reached. Junaid et al. [9] optimize outreach under fairness and time constraints, by ensuring that the expected fraction of influenced nodes in each group is the same within a prescribed time deadline. Farnadi et al. [6] propose a unifying framework that encodes all different definitions of fairness in the SIM process as constraints in a linear program that optimizes outreach. Several other works [7, 30] adopt a max-min strategy. Specifically, in Fish et al. [7] fairness is ensured by maximizing the minimum probability of a group receiving the information through modifications of the greedy algorithm. Zhu et al. [30] ensure that the outreach contains a pre-specified proportion of each group in a population. Finally, Tsang et al. [25] optimize outreach under the constraint that no group should be better off by leaving the influence maximization process with their proportional allocation of resources done internally. All these definitions involve a marginal expected value of fairness in groups, without considering the correlations – or other higher-order moments – for the joint probability distribution of different groups adopting the information (see Farnadi et al. [6] for an overview). In contrast, our work introduces a novel formalism for taking into account the actual joint distribution of outreach among groups, thus considering all groups simultaneously, highlighting limitations of various fairness metrics and developing a new seed selection policy that strategically extracts and optimizes our proposed notion of fairness. To conclude, our work is inspired by a recent line of work that draws on optimal transport theory [28] for fairness guarantees [2, 4, 21, 29, 20, 24]. To our knowledge, this is the first work to develop novel metrics and seeding algorithms that leverage optimal transport for the SIM problem.

**Motivation.** Many models of diffusion processes in the SIM problem are inherently stochastic, meaning that *who* gets the information transmitted can vary greatly from one run to another. Consider, as an example, the case in which $50\%$ of realizations over a diffusion process, no one in group 1 receives the information and everyone in group 2 does, whereas in the other $50\%$ it is the opposite. This circumstance would be classified as fair in expectation, even though it is commonly not perceived as "fair". We show how this phenomenon is common in real-world data and how our proposed framework can detect such undesired scenarios. This prompts us to study a novel fairness metric.

**Contributions.** Our main contribution is twofold: first, we propose a new fairness metric based on optimal transport, called *mutual fairness*, and second, we propose a novel seeding algorithm that optimizes for both the group-wise total outreach (termed efficiency) and fairness. Our proposed fairness metric provides stronger fairness guarantees, and it reveals and overcomes known limitations of various other fairness metrics in the literature. Specifically, we leverage optimal transport theory to build *mutual fairness*, a metric that accounts for all groups simultaneously in terms of the distance between an ideal distribution where all groups receive the information in the same proportion. We leverage our proposed mutual fairness metric to provide a unifying framework that classifies the most celebrated information-spreading algorithms both in terms of fairness and efficiency. All algorithms are tested on a variety of real-world datasets. We show how our approach unveils new insights into the role of network topology on fairness; in particular, we observe that selecting group-label blind seeds in networks with moderate levels of homophily induces inequality in information access. In contrast, very integrated or very segregated networks tend to have quite fair and efficient access to information across different groups upon greedy seedset selection. We then extend our mutual fairness metric to also account for efficiency, thus introducing the notion of $\beta$-fairness, with $\beta$ being the tuning parameter for the fairness-efficiency trade-off. Finally, we design a new seedset selection algorithm that optimizes over the proposed $\beta$-fairness metric and enhances fairness with either a small trade-off

or even improved efficiency. This novel approach provides a comprehensive evaluation and design tool that bridges the gap between fairness and efficiency in SIM problems.

## 2    Preliminaries

**Notation.** Given $m \in \mathbb{N}$, we let $[m]$ denote the interval of integers from $1$ to $m$. We denote by $G$ a network, considered undirected, and by $(C_i)_{i \in [m]}$ the $m$ groups of different sensitive attributes. In this paper, we consider $m = 2$ groups, noting that our framework is easily generalizable to more groups as discussed in Appendix B. We denote by $\phi_G(S)$ the influence function of a seedset $S$ over a network $G$, through some diffusion process. In other words, $\phi_G(S)$ determines the set of nodes reached by the seedset under a diffusion process. Then, $|\phi_G(S)|$ is often referred to as the *outreach*, a measure of efficiency for the selection of a seedset $S$. Under a stochastic diffusion process (e.g., independent cascade, linear threshold model, etc.), $|\phi_G(S)|$ is a random variable, for which we are interested in the expected value and distribution. For a particular outreach, we define the final configuration at the end of a diffusion process as follows.

**Definition 2.1 (Final configuration)** *For a network $G$ with two communities $(C_i)_{i \in [2]}$ and a seedset $S$, we let $x_i$, $i \in [2]$, denote the fraction of nodes in each community in the outreach $\phi_G(S)$. The* final configuration *is the tuple $(x_1, x_2)$.*

In many definitions in the literature, fairness is operationalized by measuring the *expected value* of the final configuration, where the expectation is taken over the diffusion process. In particular, the *equity* definition introduced by Stoica et al. [23], Junaid et al. [9] checks that the expected value of the proportions of each group reached in the outreach is the same for all groups. For a formal definition of equity and other fairness definitions in the literature, see Appendix A. We will show that relying solely on the expected value leads largely unfair outcomes to be classified as fair.

## 3    Mutual Fairness via Optimal Transport

In contrast to the literature, we propose using the *joint* outreach probability distribution, instead of its marginals, to capture simultaneous outreach between the two groups and therefore address questions like (i) When group 1 receives the information, will group 2 also receive it? (ii) Even if the two groups have the same marginal outreach probability distributions will the final configuration always be fair? We argue that capturing these aspects is crucial for understanding and assessing fairness, as shown in the motivating example below.

**Notation.** We collect the output of the information-spreading process via a probability distribution $\gamma \in \mathcal{P}([0, 1] \times [0, 1])$ over all possible final configurations. Informally, $\gamma(x_1, x_2)$ is the probability that a fraction $x_1$ of group 1 receives the information and a fraction $x_2$ of group 2 receives the information; e.g., $\gamma(0.3, 0.4)$ represents the probability that 30% of group 1 and 40% of group 2 receive the information. We can marginalize $\gamma$ to obtain the outreach probability distributions associated with each group; i.e., $\mu_1 \in \mathcal{P}([0, 1])$ and $\mu_2 \in \mathcal{P}([0, 1])$. Informally, we can write $\mu_1(x_1) = \sum_{x_2} \gamma(x_1, x_2)$. As in the example above, $\mu_i(0.3)$ is the probability that 30% of group $i$ receives the information.

**Motivating Example.** Consider the SIM problem with nodes belonging to two groups, $C_1$ and $C_2$, each group having the outreach probability distribution $\mu_i = \frac{1}{2}\delta_0 + \frac{1}{2}\delta_1, i \in \{1, 2\}$, with $\delta_k$ representing the delta distribution at $k \in [0, 1]$. That is, in 50% of the cases all members in group $i$ receive the information (i.e., we get $x_i = 1.0$) and in 50% of the cases no one in group $i$ receives the information (i.e., we get $x_i = 0.0$). It is therefore tempting to say that this setting is fair since $\mu_1$ and $\mu_2$ coincide and therefore share the same expected value. We argue that this information does not suffice to claim fairness. Indeed, consider the two following probability distributions over the final configurations:

$$\gamma_a = 0.5 \cdot \delta_{(0,0)} + 0.5 \cdot \delta_{(1,1)}, \qquad \gamma_b = 0.25 \cdot \delta_{(0,0)} + 0.25 \cdot \delta_{(1,1)} + 0.25 \cdot \delta_{(0,1)} + 0.25 \cdot \delta_{(1,0)},$$

with $\delta_{(i,j)}$, representing the delta distribution at $(i, j) \in [0, 1]^2$. Interestingly, both $\gamma_a$ and $\gamma_b$ are "compatible" with $\mu_1$ and $\mu_2$: If we compute their marginals, we obtain $\mu_1$ and $\mu_2$. However, $\gamma_a$ and $\gamma_b$ encode two fundamentally different final configurations. In $\gamma_a$, the percentage of members

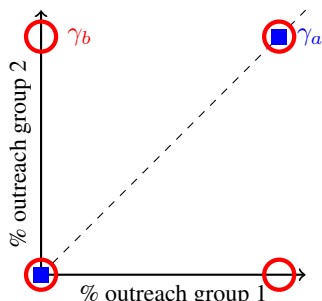
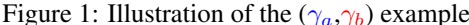
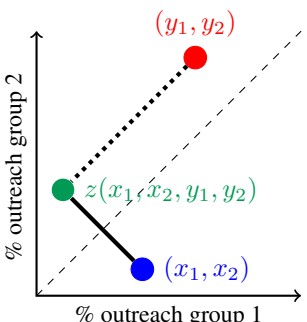

Figure 1: Illustration of the ($\gamma_a$,$\gamma_b$) example.

Figure 2: The transportation cost measures the length of the solid segment; shifts along the diagonal (dotted) are not considered for fairness and are only relevant for efficiency.

of group 1 who get the information *always* coincides with the percentage of people of group 2. Conversely, in $\gamma_b$, more outcomes are possible; in particular, there is a probability of $0.25 + 0.25 = 0.5$ that all members of one group receive the information and no member of the other group receives it (see Fig. 1). Thus, from a fairness perspective, $\gamma_a$ and $\gamma_b$ encode very different outcomes. We therefore argue that a fairness metric should be expressed in terms of *joint* probability distribution $\gamma$, and not solely based on its marginals $\mu_1$ and $\mu_2$, as commonly done in the literature [23, 9].

### 3.1 A Fairness Metric Based on Optimal Transport

Our motivating example prompts us to reason about fairness in terms of the joint probability measure $\gamma$, instead of its marginal distributions $\mu_1$ and $\mu_2$. Since $\gamma$ is a probability distribution (over all possible final configurations), we can quantify fairness by computing its "distance" from an "ideal" reference distribution $\gamma^*$ along the diagonal, capturing the ideal situation in which both groups receive the information in the same proportion. We do so by using tools from optimal transport.

**Background in optimal transport.** For a given (continuous) transportation cost $c : ([0, 1] \times [0, 1]) \times ([0, 1] \times [0, 1]) \to \mathbb{R}_{\geq 0}$, the optimal transport discrepancy between two probability distributions $\gamma_a \in \mathcal{P}([0, 1] \times [0, 1])$ and $\gamma_b \in \mathcal{P}([0, 1] \times [0, 1])$ is defined as

$$W_c(\gamma_a, \gamma_b) = \min_{\pi \in \Pi(\gamma_a, \gamma_b)} \mathbb{E}_{(x_1, x_2), (y_1, y_2) \sim \pi}, [c((x_1, x_2), (y_1, y_2))], \tag{1}$$

where $\Pi(\gamma_a, \gamma_b)$ is the set of probability distributions over $([0, 1] \times [0, 1]) \times ([0, 1] \times [0, 1])$ so that the first marginal is $\gamma_a$ and the second marginal is $\gamma_b$. Intuitively, the optimal transport problem quantifies the minimum transportation cost to morph $\gamma_a$ into $\gamma_b$ when transporting a unit of mass from $(x_1, x_2)$ to $(y_1, y_2)$ costs $c((x_1, x_2), (y_1, y_2))$. The optimization variable $\pi$ is called transportation plan and $\pi((x_1, x_2), (y_1, y_2))$ indicates the amount of mass at $(x_1, x_2)$ displaced to $(y_1, y_2)$. Thus, its first marginal has to be $\gamma_a(x_1, x_2)$ (that is, $(x_1, x_2)$ has to be transported to some $(y_1, y_2)$) and its second marginal must be $\gamma_b(y_1, y_2)$ (that is, the mass at $(y_1, y_2)$ has to arrive from some $(x_1, x_2)$). If the transportation cost $c$ is chosen to be a $p \geq 1$ power of a distance $d$, then $(W_{d^p}(\cdot, \cdot))^{1/p}$ is a distance on the space of probability distributions. When the probability distributions are discrete (or the space $[0, 1]$ is discretized), the transportation problem (1) is a finite-dimensional linear program and can therefore be solved efficiently [15].

**Our proposed fairness metric.** To operationalize the optimal transport problem (1), we therefore need to define (i) a transportation cost and (ii) a reference distribution $\gamma^*$. To define the transportation cost, we start with the following two considerations. First, moving mass *along* the diagonal should have zero cost, as it does not affect fairness but only efficiency (the proportion of population reached in respective groups). Second, moving mass orthogonally towards the diagonal should come at a price, since the difference in group proportion outreach between groups 1 and 2 decreases. We quantify this price as the Euclidean distance. This is illustrated in Fig. 2, which shows how the joint distribution captures unfairness, by depicting the percentage outreach in each group on each axis;

thus, the diagonal represents a "fair" line, where the probability of reaching a particular outreach percentage is the same for both groups.

These two insights suggest decomposing the distance between the initial configuration $(x_1, x_2)$ (e.g., belonging to $\gamma_a$) and $(y_1, y_2)$ (e.g., belonging to $\gamma_b$) into two components: one capturing efficiency and the other one being the fairness component (see Fig. 2). Since the aim of our metric is to measure fairness, we therefore obtain the transportation cost

$$c((x_1, x_2), (y_1, y_2)) = \|z(x_1, x_2, y_1, y_2) - (x_1, x_2)\| = \frac{\sqrt{2}}{2}|(x_2 - x_1) - (y_2 - y_1)|, \quad (2)$$

where $z(x_1, x_2, y_1, y_2)$ is the point indicated in green in Fig. 2 and $\|\cdot\|$ is the standard Euclidean norm. Thus, the "fairness distance" between two distributions $\gamma_a$ and $\gamma_b$ can be readily quantified by $W_c(\gamma_a, \gamma_b)$. Since moving mass along the diagonal is free, we quantify the fairness of a given $\gamma$ as its "fairness distance" from the "ideal" distribution $\gamma^* = \delta_{(1,1)}$, which represents the case where all members of both groups receive the information. We can now formally introduce our proposed fairness metric.

**Definition 3.1 (Mutual Fairness)** *Given a network with communities* $(C_i)_{i \in [2]}$, *a SIM algorithm is said to be* mutually fair *if the algorithm propagation is such that it maximizes*

$$\text{FAIRNESS}(\gamma) := 1 - \sqrt{2}W_c(\gamma, \gamma^*),$$

*where* $W_c(\gamma, \gamma^*)$ *is the optimal transport discrepancy, defined with the transportation cost* (2)*, between the probability distribution* $\gamma$ *and the desired probability distribution* $\gamma^*$ *defined as in* (1)*.*

The mutual fairness from Definition 3.1 can be seen as a normalized expression of $W_c(\gamma, \gamma^*)$ to contain its values in $[0, 1]$. Indeed, its lowest value is 0 and it is achieved with $\gamma = \delta_{(0,1)}$, for which is $W_c(\gamma, \gamma^*) = 1$; its largest value is 1 and it is achieved with $\gamma = \gamma^*$, for which $W_c(\gamma^*, \gamma^*) = 0$. Since $\gamma^*$ is a delta distribution, we can solve the optimal transport problem (1) in closed form to

$$\text{FAIRNESS}(\gamma) = 1 - \sqrt{2}W_c(\gamma, \gamma^*) = \mathbb{E}_{(x_1, x_2) \sim \gamma}\left[1 - |x_1 - x_2|\right],$$

which reduces to $\text{FAIRNESS}(\gamma) = 1 - \frac{1}{N}\sum_{i=1}^{N}|x_{1,i} - x_{2,i}|$ when the distribution $\gamma$ is empirical with $N$ samples $\{(x_{1,i}, x_{2,i})\}_{i \in [N]}$. In particular, our fairness metric can also be interpreted in terms of the average distance between the outreach proportions within the two groups.

**Discussion.** We note that, while we considered two groups in the aforementioned definition, our methodology readily extends the setting with $m$ groups. We present this extension in Appendix B. Second, since moving mass "diagonally" is free, any distribution $\gamma^*$ supported on the diagonal yields the same fairness metric. In practice, it is often not the case that all network members receive the information, and the best one could hope for is to project $\gamma$ onto the diagonal; since moving along the diagonal is free, the fairness cost is the same whether the ideal distribution is that projection or $\gamma^*$. Moreover, it is easy to see that the "fairness distance" is symmetric, namely $W_c(\gamma_a, \gamma_b) = W_c(\gamma_b, \gamma_a)$. Finally, our definition readily extends to any other distance function besides the standard Euclidean metric.

**Back to the motivating example.** Armed with a definition of fairness that captures the nature of a diffusion process, we now revisit the motivating example in Fig. 1. To start, we evaluate the "fairness distance" between $\gamma_a$ and $\gamma_b$:

$$W_c(\gamma_a, \gamma_b) = \frac{1}{4} \cdot \frac{\sqrt{2}}{2} + \frac{1}{4} \cdot \frac{\sqrt{2}}{2} = \frac{\sqrt{2}}{4},$$

which amounts to the cost of transporting the points $(0, 1)$ and $(1, 0)$, each with weight $1/4$, to the diagonal. Notably, in contrast to simply computing the expected outreach of each group, our fairness metric distinguishes the two outcomes. Similarly, we can easily compute the fairness metric: $\text{FAIRNESS}(\gamma_a) = 1$ and $\text{FAIRNESS}(\gamma_b) = 0.5$. In particular, $\gamma_a$ achieves the highest fairness score. Indeed, its outcome will always be fair. Instead, $\text{FAIRNESS}(\gamma_b)$ achieves a lower fairness score, capturing the fact that in 50% of the cases the outcome is perfectly fair, while in the remaining 50% it is largely unfair.

## 3.2 Mutual Fairness in Practice

We now investigate the use of our newly defined fairness metric across a variety of real-world datasets: Add Health (`AH`), Antelope Valley variants 0 to 23 (`AV_{0-23}`) [26], APS Physics (`APS`) [13], Deezer (`DZ`) [19], High School Gender (`HS`) [14], Indian Villages (`IV`) [1], and Instagram (`INS`) [22]. Each dataset contains a social network with a chosen demographic partitioning the population into two groups (see Appendix C for details). We load the datasets as graphs $G(V, E)$. We then select a seedset $S$ of size 2-90 (depending on the dataset) using the following heuristics: two group-agnostic seed selection strategies as our baselines, namely *degree centrality* (`bas_d`), and *greedy* (`bas_g`), proposed by Kempe et al. [11]. In addition, we implement two fair seed selection heuristics based on the equity metric, namely *degree-central fair heuristic* (`hrt_d`), and *greedy fair heuristic* (`hrt_g`), proposed by Stoica et al. [23]. To model the information spread, we use the Independent Cascade model (`IC`) for the diffusion of information [11] with a probability $p \in [0, 1]$ for all edges. This process, being stochastic, is simulated $R$ times in a Monte Carlo sampling process to achieve $R$ *final configurations* (Definition 2.1) plotted together as a *joint outreach distribution*, in Fig. 3. Then we apply our distribution-aware notion of fairness from Section 3.1, mutual fairness. We keep $R = 1,000$ throughout, but explore several values in $p, |S|$ (mentioned per experiment in the figures below) and exhaustively recorded with other hyperparameters in Appendix D. All details related to computational resources and development environment are available in Appendix G. The code for all our numerical experiments is available at https://github.com/nicolaslanzetti/fairness-sim-ot.

**Are the outcomes fair?** As a first experiment, we study the *joint* outreach probability distribution for different datasets. We identify four qualitatively different outcomes, shown in Fig. 3 for a few of the datasets. Additional experiments with different propagation probability and seed selection strategies can be found in Appendix D. Fig. 3a is obtained on `AH` with `bas_g` selection strategy and $p = 0.5, |S| = 10$. We note how the joint outreach distribution is almost *concentrated* on the top right of the plane, i.e., the outcome is almost *deterministic* and highly fair and efficient. In turn, this trivializes both the expected value in the equity metric and the cost in the mutual fairness metric in Definition 3.1, which therefore essentially boils down to comparing the almost deterministic outreach fraction within each group. In these cases, our fairness metric does not provide additional insights. Such deterministic outcomes are typical of degree or greedy seedset outreach in dense graphs, such as `AH, DZ, INS` (refer to Appendix D), with extreme probability of conduction ($p \geq 0.5$ or $p \to 0$), and cross-group interconnectivity (see Table 1 in Appendix C). For moderate $p$ (e.g., $0.1$), the outreach probability distribution is concentrated along the diagonal (Fig. 3b). Thus, both the equity metric and our fairness measure are *maximal*. Nonetheless, our fairness metric provides additional insights: not only does the expected outreach within each group coincide, but also the outreach at *every* realization coincides (see the example in Section 3). Thus, our fairness metric provides a stronger certificate of fairness. As before, the same applies to `AH, DZ, INS` (see Appendix D). Intuitively, high cross-group interconnectivity in a dense graph already ensures fairness. Additionally, extreme $p$ values ensure deterministic outreach (either the information dies out at the seedset, or reaches everyone in the population). When propagation happens with moderate propagation probabilities, $p$, outreach appears as in Fig. 3b. Fig. 3c represents `APS` for its `hrt_g` seedset outreach and $p = 0.3, |S| = 6$. Here, we observe a highly stochastic outcome, with many realizations for which almost no member of one group receives the information. Note that the phenomenon observed in Fig. 3c is the same as the one captured by our motivating example. We argue such an outcome should *not* be classified as fair, despite the expected value of the proportions being similar. Finally, Fig. 3d shows the `AV_0` dataset with $p = 0.3, |S| = 4$, and `bas_g` selection strategy. We observe a more stochastic outreach compared to Fig. 3b with variance spread along, but not on the diagonal, with a small bias towards one group. Also in this case, both the equity and mutual fairness metrics characterize this outcome as fair, but mutual fairness is more informative as it requires outcomes to be fair at each realization.

**The impact of the conduction probability.** As a second experiment, we investigate the difference between mutual fairness and equity (difference in the expected value of the proportions), as a function of the conduction probability $p$. We consider the `IV` dataset as a case study and select seeds using `bas_g`. We show our results in Fig. 4. Our mutual fairness metric in Definition 3.1 shows a fundamentally different trend compared to the equity metric from Definition A.3. Importantly, for $p \in (0, 0.5)$, both metrics have an opposite trend: equity fairness increases to some extent whereas our metric suggests a significant fall in fairness in this region. For $p \in (0.5, 0.7)$, there is a decrease in equity fairness, while our fairness evaluation remains relatively constant. We notice similar trends

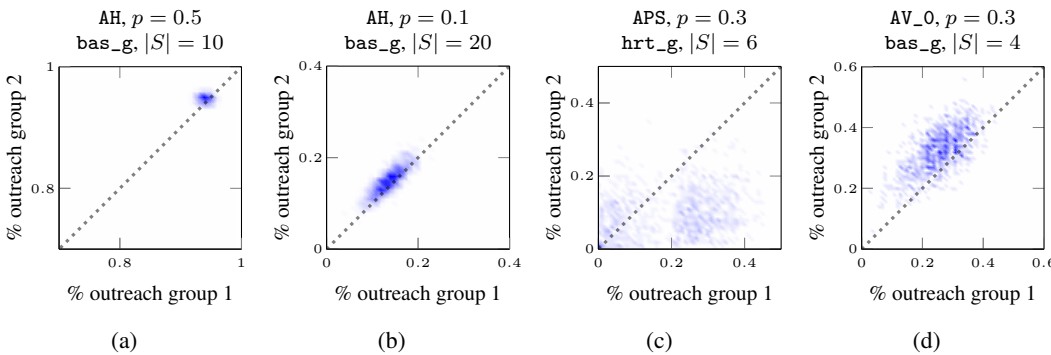

Figure 3: Joint outreach probability distribution for different datasets, different propagation probabilities $p$, and seedsets cardinalities $|S|$.

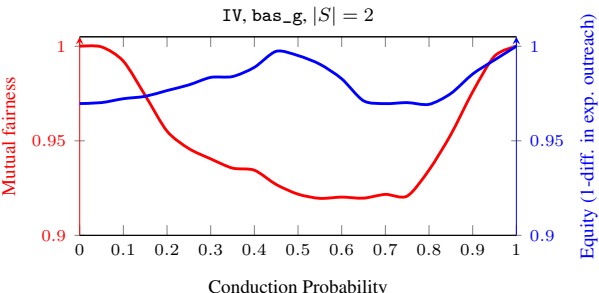

Figure 4: Mutual fairness (left, red) and equity (right, blue) for the IV dataset as $p$ varies in $[0, 1]$.

for both metrics only for $p \in (0.8, 1.0)$. The significant difference in the trend of the two metrics confirms our previous finding that mutual fairness is more informative than the equity metric and that the equity metric fails to adequately capture changes in fairness, see Sections 3.1 and 3.2. For more experiments on other datasets, we refer to Appendix D.2.

### 3.3  Trading off Fairness and Efficiency

To construct our fairness metric, we completely discarded the efficiency of the final configuration. For instance, the "fairness distance" between a configuration whereby no agent receives the information (i.e., $\gamma = \delta_{(0,0)}$) and the "ideal" configuration whereby everyone receives the information (i.e., $\gamma^*$) is zero, as both probability distributions lay on the diagonal. As such, the fairness score of $\gamma = \delta_{(0,0)}$ is 1 and therefore maximal. Thus, in practice, one seeks a fairness-efficiency *tradeoff*.

In our setting, we can easily introduce the tradeoff in the transportation cost (2). Specifically, we can define the transportation cost as a weighted sum of the "diagonal distance" (measuring the difference in efficiency, dotted segment in Fig. 2) and the "orthogonal distance" (measuring the difference in fairness, solid segment in Fig. 2). Formally, for a given weight $\beta \geq 0$, the transportation cost reads

$$c_\beta((x_1, x_2), (y_1, y_2)) = \beta \|z(x_1, x_2, y_1, y_2) - (x_1, x_2)\| + (1 - \beta)\|z(x_1, x_2, y_1, y_2) - (y_1, y_2)\|$$

$$= \beta \frac{\sqrt{2}}{2}|(x_2 - x_1) - (y_2 - y_1)| + (1 - \beta)\frac{\sqrt{2}}{2}|(x_1 + x_2) - (y_1 + y_2)|. \quad (3)$$

We refer to Fig. 5 for a heatmap of $c_\beta$. In particular, for $\beta = 1$, we recover the transportation cost (2); for $\beta = 0$ one optimizes for efficiency, and the $\beta$-fairness collapses in the classical influence maximization problem. We can then proceed as in Section 3.1. The "$\beta$-fairness-efficiency distance" between $\gamma_a$ and $\gamma_b$ is $W_{c_\beta}(\gamma_a, \gamma_b)$ and the $\beta$-fairness metric can be then defined as follows.

**Definition 3.2** ($\beta$-**Fairness**) *Consider a network with groups $C_1, C_2$, a SIM algorithm is said to be $\beta$-fair if the algorithm propagation is such that it maximizes*

$$\beta-\text{FAIRNESS}(\gamma) := 1 - \frac{\sqrt{2}}{\max\{1, 2 - 2\beta\}} W_{c_\beta}(\gamma, \gamma^*), \quad (4)$$

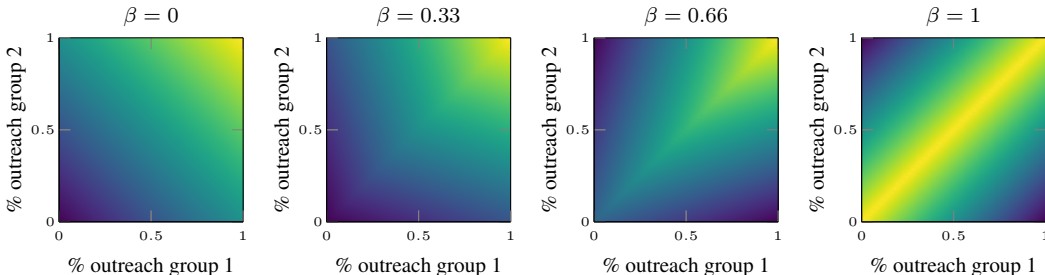

Figure 5: Cost of transporting a point $(x_1, x_2)$ to the "ideal" point $(1, 1)$ (i.e., everyone receives the information) for various values of $\beta$ (i.e., we plot $(x_1, x_2) \mapsto c_\beta((x_1, x_2), (1, 1))$). Yellow denotes a low transportation cost, whereas dark blue denotes a large cost.

*with $W_{c_\beta}(\gamma, \gamma^*)$ defined as in (1) with transportation cost as in (3) and ideal distribution $\gamma^* = \delta_{(1,1)}$.*

The terms $1$ and $\sqrt{2}/\max\{1, 2 - 2\beta\}$ in (4) ensure that the metric is non-negative and in $[0, 1]$. Again, the optimal transport problem can be solved in closed form, which yields

$$\beta-\text{FAIRNESS}(\gamma) = \mathbb{E}_{(x_1, x_2) \sim \gamma} \left[ 1 - \frac{\beta|x_1 - x_2| + (1 - \beta)|x_1 + x_2 - 2|}{\max\{1, 2 - 2\beta\}} \right].$$

In particular, for $\beta = 1$, we recover the mutual fairness $\text{FAIRNESS}(\gamma)$ in Definition 3.1 and for $\beta = 0$ we obtain the efficiency metric $\mathbb{E}_{(x_1, x_2) \sim \gamma}[1 - \frac{|x_1 + x_2 - 2|}{2}]$.

## 4   Improving Fairness

### 4.1   Fairness-promoting Seed-selection Algorithm

Armed with a novel fairness metric, $\beta-\text{FAIRNESS}$, we now design an *iterative* seed-selection algorithm, which we call *Stochastic Seedset Selection Descent* (S3D), that strategically selects seeds taking into account all communities simultaneously. The pseudo-code is summarized in Algorithm 1. For its motivation and details, refer to Appendix E. For a given initial seedset, our algorithm explores new seeds and evaluates them on the efficiency-fairness metric $\beta-\text{FAIRNESS}$ as in (4) for a desired value of the fairness-efficiency tradeoff parameter $\beta$ (S3D_STEP() in Appendix E), to decide if the new seedset becomes a candidate for the optimized seedset. These seeds are searched for by iteratively sampling stochastically reachable nodes, up to a fixed depth, taken as a fraction of the graph diameter, from the current seedset (SEEDSET_REACH() in Appendix E) while making sure they contribute to a non-overlapping outreach (Algorithm 1::6-8). To avoid local minima of the generally non-convex objective, the procedure allows for visiting inferior seedsets on $\beta-\text{FAIRNESS}$ or even selecting completely random ones on rare occasions (Algorithm 1::12-18) using *Metropolis Sampling* [18]. Otherwise, a high $\beta-\text{FAIRNESS}$ encourages opting for the new seedset with high probability. Finally, we revisit all the seedset candidates collected so far and pick the one with the largest $\beta-\text{FAIRNESS}$ as the optimal seedset. For a sparse graph $G(V, E)$, with $E = O(V)$, choosing $|S|$ seeds, averaging over $R$ realizations to approximate outreach via Monte-Carlo sampling and exploring $k$ candidates using S3D_STEP suggests a total running time upper bound of $O(kR|S||V|)$ (see Appendix E for details). In practice, $k \in [500, 1000], R = 1000$ for $S \in [2, 100]$ works well for all datasets.

### 4.2   Real-world Data

**Are the outcomes more fair?**   We test our algorithm across a variety of datasets (Appendix C) against our baselines (bas_d, bas_g). We initialize the S3D algorithm with the two baseline seedsets and hence include results from two separately optimized seedsets, S3D_d, S3D_g. Our results are shown in Fig. 6. Informally, we observe that our seed-selection mechanism "moves" the probability mass of the joint outreach probability distribution towards the diagonal, which ultimately increases the fairness of the resulting configuration. At the same time, efficiency either increases as well or suffers only a small decrease, as we investigate more in detail in our next experiment. Generally, datasets

---

**Algorithm 1** Stochastic Seedset Selection Descent

---

**Input**: Social Graph $G(V_G, E_G)$, initial seed set $S_0$, $\beta$ fairness weight, $\epsilon$-tolerance
**Output**: Optimal seedset $S^*$

 1: $\mathcal{S} \leftarrow \{\}$, $S \leftarrow S_0$               ▷ initial collection of candidates, running seedset
 2: **for** $k$ iterations **do**                      ▷ configurable $k$
 3:      $V_S \leftarrow$ nodes reachable from $S$ via cascade, using SEEDSET_REACH routine
 4:      $S' \leftarrow \{\}$
 5:      **for** $|S|$ iterations **do**      ▷ searching nearby states, $V_{S'}$, to get $S'$ (Appendix E.3)
 6:          $S' \leftarrow S' \cup \{v\} \mid v \sim V_S$
 7:          $V_{S'} \leftarrow$ nodes reachable from $S'$ in a fixed horizon, using SEEDSET_REACH
 8:          $V_S \leftarrow V_S \setminus V_{S'}$
 9:      $E_S \leftarrow -\text{BETA\_FAIRNESS}(S, \beta)$      ▷ expected potential energy defined on $\beta$-fairness
10:      $E_{S'} \leftarrow -\text{BETA\_FAIRNESS}(S', \beta)$
11:      $p_{\text{accept}} \leftarrow \min\{1, e^{E_S - E_{S'}}\}$       ▷ $S'$ acceptance on energy minimization
12:      **if** $x \sim \mathcal{B}(p_{\text{accept}})$ **then**          ▷ Metropolis sampling
13:          $S^+ \leftarrow S'$                  ▷ get a better seedset
14:      **else**
15:          **if** $x \sim \mathcal{B}(\epsilon)$ **then**          ▷ for some small constant $\epsilon$
16:              $S^+ \leftarrow \{v_i\}_{i=1}^{|S|} \overset{|S|}{\sim} V_G$       ▷ random seedset
17:          **else**
18:              $S^+ \leftarrow S$            ▷ retain existing choice
19:      $\mathcal{S} \leftarrow \mathcal{S} \cup \{S^+\}$
20:      $S \leftarrow S^+$                ▷ for next iteration
21: $S^* \leftarrow S \in \mathcal{S} \mid \text{BETA\_FAIRNESS}(S, \beta)$ is maximum      ▷ via S3D_ITERATE
22: **return** $S^*$

---

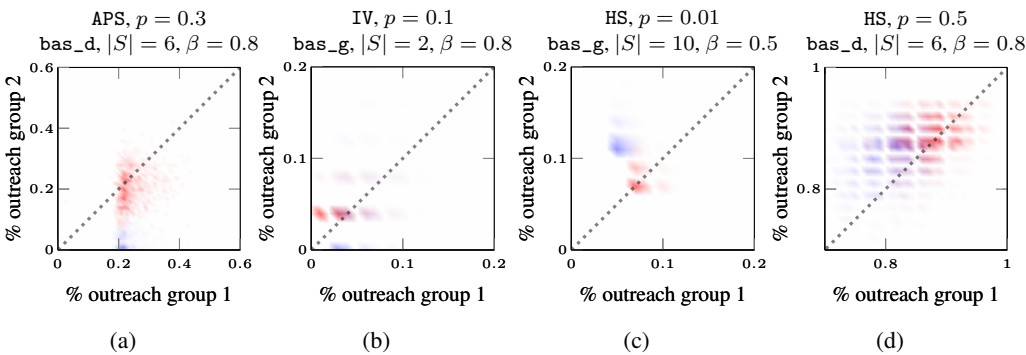

(a)             (b)             (c)             (d)

Figure 6: Demonstrate S3D (red) improvement over its label-blind baseline counter-part initializations (blue) for several datasets, propagation probabilities $p$, seed set cardinalities $|S|$ and fairness-efficiency tradeoffs $\beta$. Fig. 6d provides the strongest evidence that, besides improving in fairness, our strategy can also be more efficient, from $83.1\%$ to $87.9\%$.

with high cross-group connections (AH, DZ, INS) yield moderately fair outreach with label-blind seed selection. Similarly, for datasets with low cross-group connections (APS) a label-blind strategy, in order to maximize efficiency, selects a diverse population of seeds from which all communities are reached. Therefore, label-blind algorithms work similarly to S3D. In other moderate cases (AV, HS, IV), instead, we observe significant improvements of S3D over label-blind strategies.

**Classification of seed-selection algorithms.** In our final experiments, we compare several algorithms along with ours in terms of efficiency and mutual fairness across various datasets (see Appendix C). We consider the following algorithms: bas_d, bas_g, their fair heuristic counterparts, hrt_d, hrt_g, against our S3D_d, S3D_g, initialized via greedy and degree centrality baseline seeds, respectively. We show our results in Fig. 7. S3D achieves in almost all cases the highest

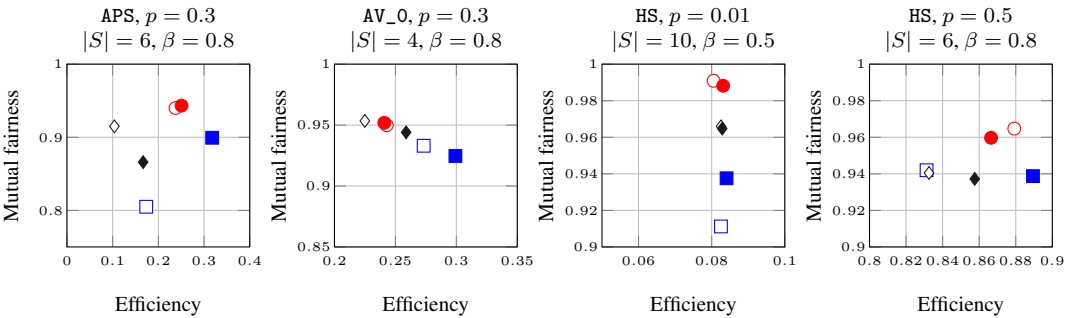

Figure 7: `S3D` trade-off and improvement against other label-aware and label-blind algorithms for several datasets, propagation probabilities $p$, seed set cardinalities $|S|$ and fairness-efficiency tradeoffs $\beta$. Filled markers refer to greedy-based algorithms: $\blacksquare = $ `bas_g`, $\bullet = $ `S3D_g`, and $\blacklozenge = $ `hrt_g`. Empty markers refer to degree-based algorithms: $\square = $ `bas_d`, $\circ = $ `S3D_d`, and $\lozenge = $ `hrt_d`. For statistical bounds, we refer to Appendix F.

fairness score ($y$-axis) and generally a slightly lower efficiency score ($x$-axis), compared to others. Thus, our seed-selection mechanism leads to fairer outcomes with only a minor decrease in efficiency.

**The impact of the network topology.**    To conclude, we discuss the impact of the network topology. In particular, when the conduction probability is moderate, network topology starts playing a role, mainly through the number of cross-group edges (CE):

*CE% is small ($\sim$ 5%, `APS`):* Such datasets encode group interaction information in the edges themselves, that is, an edge likely means nodes belong to the same group. In such cases, baseline greedy algorithms (`bas_g`) already perform well as they rely only on edge connectivity. In such circumstances, `S3D` does not significantly improve on their selection, both in efficiency and fairness.

*CE% is balanced (40-50%, `HS`, `AH`):* These datasets reflect that groups interact well across each other and so any seedset selection largely ends up in a fair outreach. Since `bas_g` already has proven near-optimal efficiency guarantees, it is unlikely that `S3D` performs significantly better than `bas_g`.

*CE% is moderate (5-30%, `AV` (datasets 0, 2, 16, 20), `IV`):* These are the non-trivial cases not covered above. Here `bas_g` can not reliably leverage the existence of edges into group information. Hence, `S3D` usually outperforms the baseline, achieving similar efficiency scores while significantly improving fairness.

*CE% is high (>50%):* The case where nodes interact more across groups than in their group was never observed. However, as long as the existence of edges does not reliably signal group information, we expect `S3D` to perform well based on a similar analysis.

*Moderate outreach in dense graphs (`INS`, `DZ`):* For graphs where $|E|$ substantially exceeds $|V|$, the outreach variance across sample sub-graphs is too low to be captured in the discretized space we experimented ($100 \times 100$ units in $[0, 1]^2$), even for moderate $p$. This leads to single-point concentrated joint-distribution plots, all of them leading to the same $\beta-$FAIRNESS.

## 5   Conclusions and Limitations

**Conclusions.**    We propose a new fairness metric, called mutual fairness, in the context of SIM. Mutual fairness draws on optimal transport and captures various fairness-related aspects (e.g., when members of group 1 receive the information will members of group 2 receive it?) that are obscure to the fairness metrics in the literature. We also leverage our novel fairness metric to design a new seed selection strategy that tradeoffs fairness and efficiency. Across various real datasets, our algorithm yields superior fairness with a minor decrease (and in some cases even an increase) in efficiency.

**Limitations.**    Our proposed algorithm, `S3D`, is essentially a random combinatorial search in the graph defining the social network. As such, its performance will generally depend on the quality of the seedset initialization. Moreover, there is no guaranteed bound on the number of iterations needed in `S3D` to achieve a desired level of fairness. Both aspects can be limiting in real-world applications.

## Acknowledgments and Disclosure of Funding

We thank the reviewers for their constructive suggestions. This work was supported as a part of NCCR Automation, a National Centre of Competence in Research, funded by the Swiss National Science Foundation (grant number 51NF40_225155). A.-A. S. acknowledges support from the Tübingen AI Center.

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

# A    Existing Fairness Metrics

**Definition A.1 (Expected outreach ratio)**  *Given a network with communities $C_1, \ldots, C_m$, the SIM algorithm* expected outreach ratio in $C_i$, $\bar{x}_i$, *is the expected ratio of nodes reached in $C_i$, namely*

$$\bar{x}_i := \frac{\mathbb{E}[|\{v \text{ reached } |v \in C_i\}|]}{|C_i|}, \quad \forall i \in \{1, \ldots, m\}.$$

**Definition A.2 (Equality [23])**  *Given the groups $C_1, \ldots, C_m$, a configuration is said to be* equal, *if the SIM algorithm chooses a seed set $S$ in a way such that the proportion of all communities in the seed set is the same, namely*

$$\frac{\mathbb{E}[|\{v \in S|v \in C_i\}|]}{|C_i|} = \frac{\mathbb{E}[|\{v \in S|v \in C_j\}|]}{|C_j|} \quad \forall i, j \in \{1, \ldots, m\}.$$

The notion of equality focuses on the fair allocation of seeds to the groups proportional to the size of the group within the population. This notion of fairness applies, for example, in the context of advertising companies that aim at having a fair distribution of resources among groups.

**Definition A.3 (Equity [23])**  *Given a network with communities $C_1, \ldots, C_m$, a SIM algorithm that selects a seedset $S$ is said to be* equitable *if the algorithm propagation reaches all communities in a balanced way, i.e. $\bar{x}_i = \bar{x}_j$ for all $i, j \in \{1, \ldots, m\}$.*

The notion of equity focuses on the outcome of the diffusion process, e.g. independent cascade, linear threshold model and it is suitable in contexts in which one aims to reach a diverse population in a calibrated way.

**Definition A.4 (Max-min fairness [6])**  *Given the groups $C_1, \ldots, C_m$, the* max-min fairness *criterion maximizes the minimum expected outreach ratio among all groups, namely* $\max \min_{i \in \{1, \ldots, m\}} \bar{x}_i$.

The goal of the maxmin fairness is to minimize the gap among different groups in the outreach. The SIM problem under maxmin constraints has been investigated in [6, 7, 30].

**Definition A.5 (Diversity [6])**  *Given the groups $C_1, \ldots, C_m$, let $k_i = \left\lceil k \cdot \frac{|C_i|}{|V|} \right\rceil$, where $k$ is the pre-specified total seed budget. Let $\bar{x}_i^*(C_i) := \max_{S \subset C_i : |S| = k_i} \bar{x}_i$. A configuration is said to be* diverse *if for each $i \in \{1, \ldots, m\}$ it holds $\bar{x}_i \geq \bar{x}_i^*(C_i)$, where $\bar{x}_i$ refers to the expected outreach ratio in $C_i$ obtained from the seed set S, with $|S| = k$.*

The notion of diversity ensures that each group receives influence at least equal to their internal spread of influence. The SIM problem under diversity constraints has been investigated in [6, 25].

# B    Extension to Multiple Groups

In this section, we extend our definitions of mutual fairness and $\beta$-fairness to the setting of $m$ groups. To do so, we first notice that, in the case of $m$ groups, the outreach distribution is a probability distribution $\gamma$ on the hypercube $[0, 1]^m$; i.e., $\gamma$ now lives in $\mathcal{P}([0, 1]^m)$.

We start with the definition of mutual fairness. We proceed as in Section 3.1 and define mutual fairness via optimal transport, which, in turn, requires defining a reference distribution and a transportation cost. The reference distribution is again the "ideal" distribution $\gamma^* = \delta_{(1, \ldots, 1)}$ which encodes the case in which all members of all groups receive the information. As for the transportation cost, it suffices to generalize the transportation cost (2) to an $m$ dimensional space. Specifically, it can be defined as the distance between any given point $(x_1, \ldots, x_m) \in [0, 1]^m$ in the hypercube and the diagonal line. For this, let

$$z(x_1, \ldots, x_m) = \underset{z=(y,\ldots,y), y \in [0,1]}{\operatorname{argmin}} \|(x_1, \ldots, x_m) - z\|$$

$$= \frac{x_1 + \ldots + x_m}{m}(1, \ldots, 1)$$

be the closest point to $(x_1, \ldots, x_m)$ on the diagonal. Then, the transportation cost can be defined as in (2) and the fairness metric reads

$$\textsc{Fairness}(\gamma) = 1 - \alpha \mathbb{E}_{(x_1,\ldots,x_m)\sim\gamma}[\|z(x_1,\ldots,x_m) - (x_1,\ldots,x_m)\|]$$

$$= 1 - \alpha \mathbb{E}_{(x_1,\ldots,x_m)\sim\gamma}\left[\min_{z\in[0,1]} \|(x_1,\ldots,x_m) - (z,\ldots,z)\|\right],$$

where the constant $\alpha > 0$ is again chosen so that $\textsc{Fairness}(\gamma)$ is between 0 and 1. Note that in the case of two groups, we have $z = \frac{1}{2}(x_1 + x_2)$ and

$$\min_{z\in[0,1]} \|(x_1,\ldots,x_m) - (z,\ldots,z)\| = \frac{\sqrt{2}}{2}|x_1 - x_2|,$$

which is precisely the mutual fairness of Definition 3.1.

We now turn our attention to $\beta$-fairness. We can proceed analogously and obtain

$$\beta-\textsc{Fairness}(\gamma) = 1 - \alpha \mathbb{E}_{(x_1,\ldots,x_m)\sim\gamma}[\beta\|z(x_1,\ldots,x_m) - (x_1,\ldots,x_m)\| + (1-\beta)\|z(x_1,\ldots,x_m) - (1,\ldots,1)\|],$$

where $\alpha > 0$ is again chosen to normalize the metric. Again, in the case of two groups, we have $z = \frac{1}{2}(x_1 + x_2)$ and so

$$\beta-\textsc{Fairness}(\gamma) = 1 - \alpha \mathbb{E}_{(x_1,\ldots,x_m)\sim\gamma}\left[\beta\frac{\sqrt{2}}{2}|x_1 - x_2| + (1-\beta)\frac{\sqrt{2}}{2}|x_1 + x_2 - 2|\right],$$

which coincides with Definition 3.2.

We conclude with two remarks on this extension to $m$ groups. First, as in the case of two groups, there is no need to numerically solve optimal transport problems, as we provide a closed-form expression for the optimal transport problems. Second, we highlight that our extension to $m$ groups does not resort to the so-called multi-marginal optimal transport problem, which might cause exponential complexity in the dimensionality.

## C  Description and Properties of Datasets

To associate the notion of fairness developed in Sections 3.1 and 3.3 with the datasets and the outcomes from experiments in Sections 3.2 and 4.2, we summarize the dataset statistics in Table 1. *Minority %* is calculated as the percentage of the minority group nodes in the entire population. *Fraction of Cross Edges* evaluates *heterophily* in the dataset, by calculating the fraction of edges that connect different groups. A higher value means a more heterophilic network, whereas a lower value means a more homophilic network.

**Add Health (AH).**  The Add Health dataset consists of a social network of students in schools and a relation between them is represented by whether they nominated each other in the Add Health surveys. We select a school at random with $1,997$ students and use race as the sensitive attribute (white and non-white).[2]

**Antelope Valley (AV), [26].**  We choose 4 random networks among the 24 available in the Antelope Valley dataset to compare our fairness-improving algorithm, S3D, against [26], which worked on the same dataset. We also run our baselines and other fair seed selection heuristics from [23] on these datasets to get a fair comparison. The two sensitive attribute groups are male and female, self-reported in the dataset with binary attributes.

---

[2]The Add Health project is funded by grant P01 HD31921 (Harris) from the Eunice Kennedy Shriver National Institute of Child Health and Human Development (NICHD), with cooperative funding from 23 other federal agencies and foundations. Add Health is currently directed by Robert A. Hummer and funded by the National Institute on Aging cooperative agreements U01 AG071448 (Hummer) and U01AG071450 (Aiello and Hummer) at the University of North Carolina at Chapel Hill. Add Health was designed by J. Richard Udry, Peter S. Bearman, and Kathleen Mullan Harris at the University of North Carolina at Chapel Hill.

| Dataset | # Nodes | # Edges | Avg. Degree | Diameter | Minority % | Frac. Cross Edges |
|---------|---------|---------|-------------|----------|------------|-------------------|
| AH      | 1997    | 8523    | 8.54        | 10       | 34.6       | 0.452             |
| AV_0    | 500     | 969     | 3.87        | 12       | 49         | 0.189             |
| AV_2    | 500     | 954     | 3.81        | 14       | 49.6       | 0.183             |
| AV_16   | 500     | 949     | 3.8         | 13       | 47.6       | 0.210             |
| AV_20   | 500     | 959     | 3.84        | 15       | 48.4       | 0.198             |
| APS     | 1281    | 3064    | 4.78        | 26       | 31.8       | 0.056             |
| DZ      | 18442   | 46172   | 5.00        | 25       | 44.4       | 0.476             |
| HS      | 133     | 401     | 6.03        | 10       | 40.6       | 0.394             |
| IV      | 90      | 238     | 5.29        | 13       | 26.7       | 0.265             |
| INS     | 553628  | 652830  | 2.36        | 16       | 45.6       | 0.417             |

Table 1: Summary statistics of datasets used.

**APS Physics (APS), [13].** The APS citation network contains $1,281$ nodes, representing papers written in two main topics: Classical Statistical Mechanics (CSM), constituting $31.8\%$ of the papers, and Quantum Statistical Mechanics (QSM), accounting for the rest. As Lee et al. [13] analyze, the dataset has high homophily, meaning that each subfield cites more papers in its own field than in the other field. For simplicity, we use only the largest connected component in the full dataset (component stats in 1) between the two groups, for this study.

**Deezer (DZ), [19].** A social network from Europe with $18,442$ nodes, where each node has a self-reported attributed gender (male or female). Men are the minority ($44.3\%$) and women are the majority ($55.6\%$). The dataset has moderate homophily.

**High School (HS), [14].** A high school friendship network collected from Mastrandrea et al. [14], with 133 nodes in its main connected component represented by students who self-identify as male or female. The majority are female ($60\%$), and the network is homophilic.

**Indian Villages (IV), [1].** The dataset contains different demographic attributes for the individual networks and the household networks collected in 77 Indian villages, from which we select Mother-tongue (Telugu or Kannada) as the sensitive attribute. We note that most villages contain a majority mother tongue, either Telugu or Kannada. We pick a random village with 90 individuals for our study.

**Instagram (INS), [22].** An interaction network from Instagram containing $553,628$ nodes, where everyone has a labeled gender ($45.57\%$ men and $54.43\%$ women). Each edge between two users represents a 'like' or 'comment' that one user gave another on a posted photo. The dataset has moderate homophily.

# D   Details on the Experiments and Extended Results

We use $R = 1000$ throughout our experiments. For the outreach, we discretize the space $[0,1] \times [0,1]$ into $100 \times 100$ equal sized bins. For S3D (refer to Appendix E), we use constants, `exploit_to_explore` $= 1.3$, `non_acceptance_retention_prob` $= 0.95$, and `shallow_horizon` $= 4$.

## D.1   Outreach Distribution

We report additional experiments in Figs. 8 to 11.

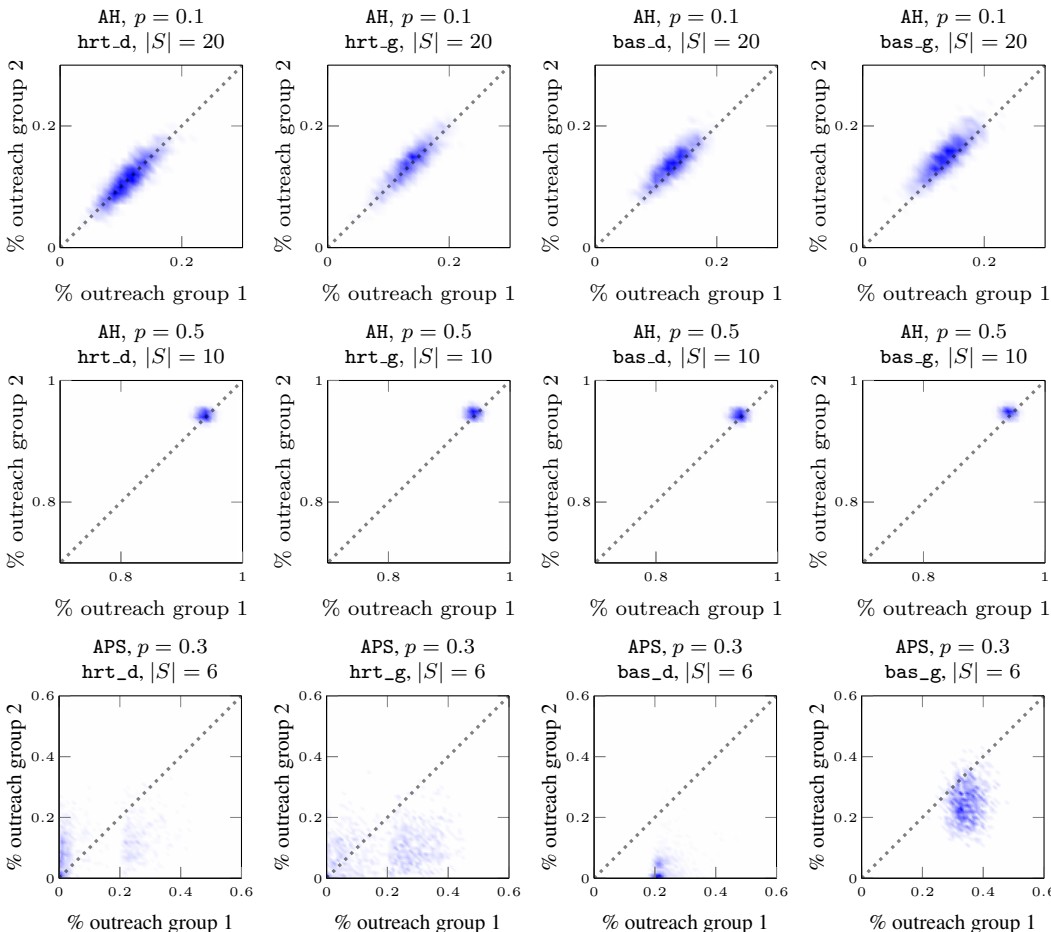

Figure 8: Outreach distribution.

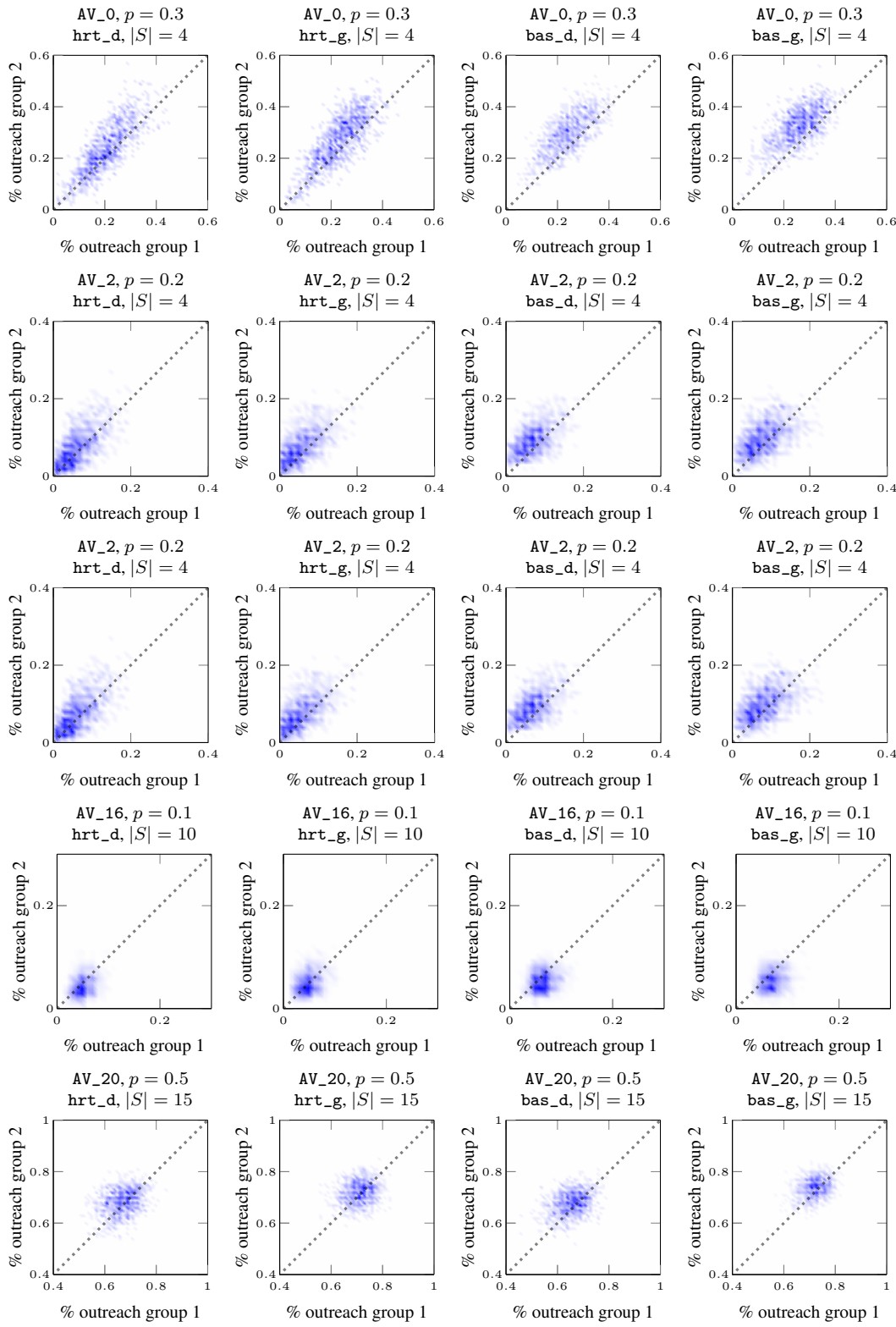

Figure 9: Outreach distribution.

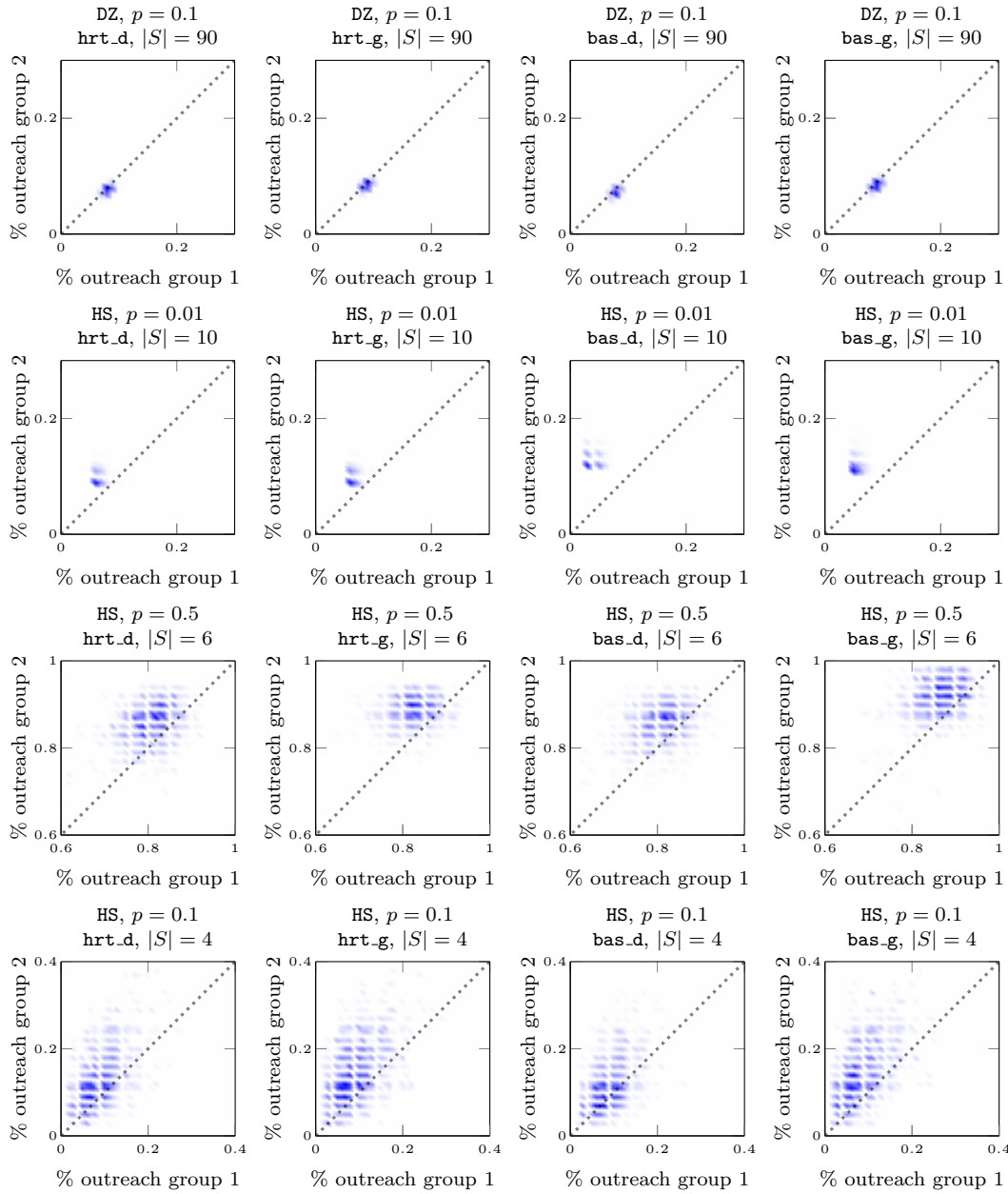

Figure 10: Outreach distribution.

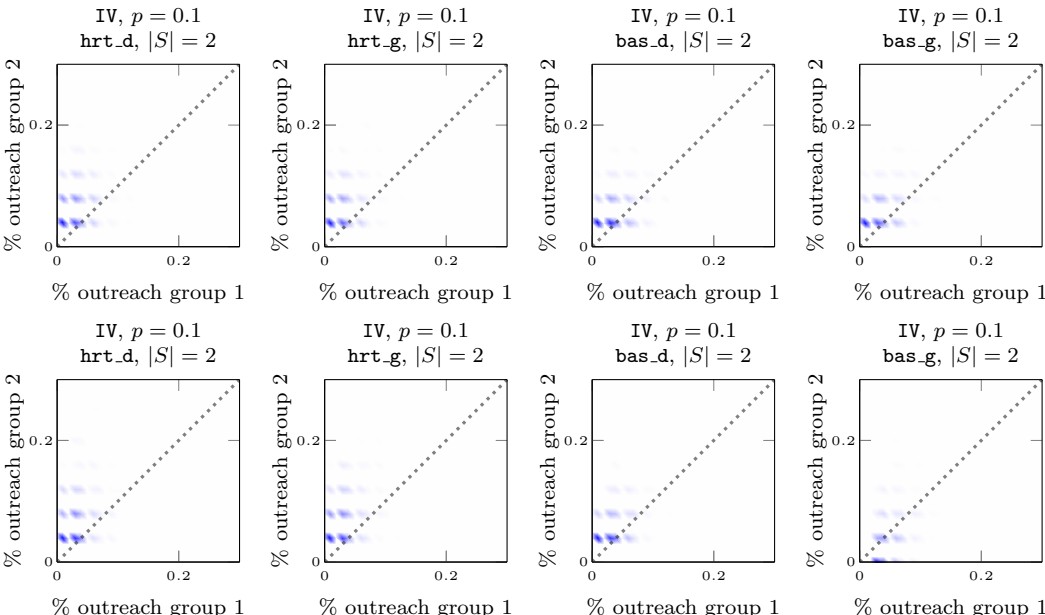

Figure 11: Outreach distribution.

## D.2 The Impact of the Conduction Probability for Various Dataset

We report additional experiments in Figs. 12 and 13.

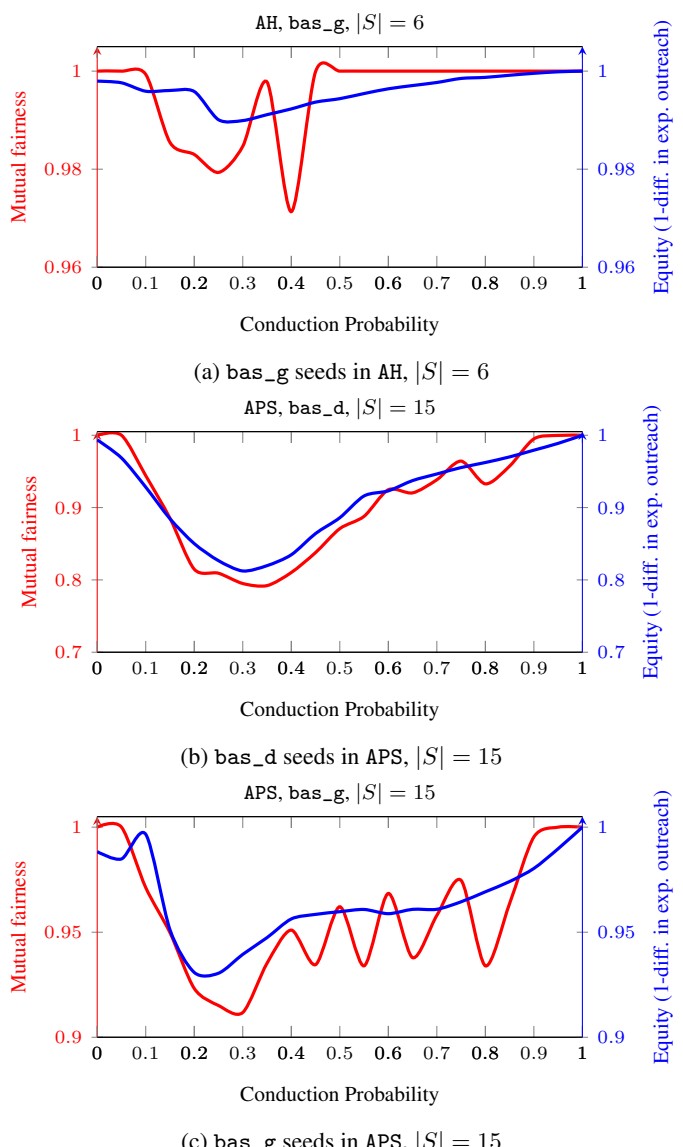

(a) `bas_g` seeds in `AH`, $|S| = 6$

(b) `bas_d` seeds in `APS`, $|S| = 15$

(c) `bas_g` seeds in `APS`, $|S| = 15$

Figure 12: Part 1: Different definitions of fairness VS conduction probability on an outreach distribution created by the `bas_g` or `bas_d` heuristic.

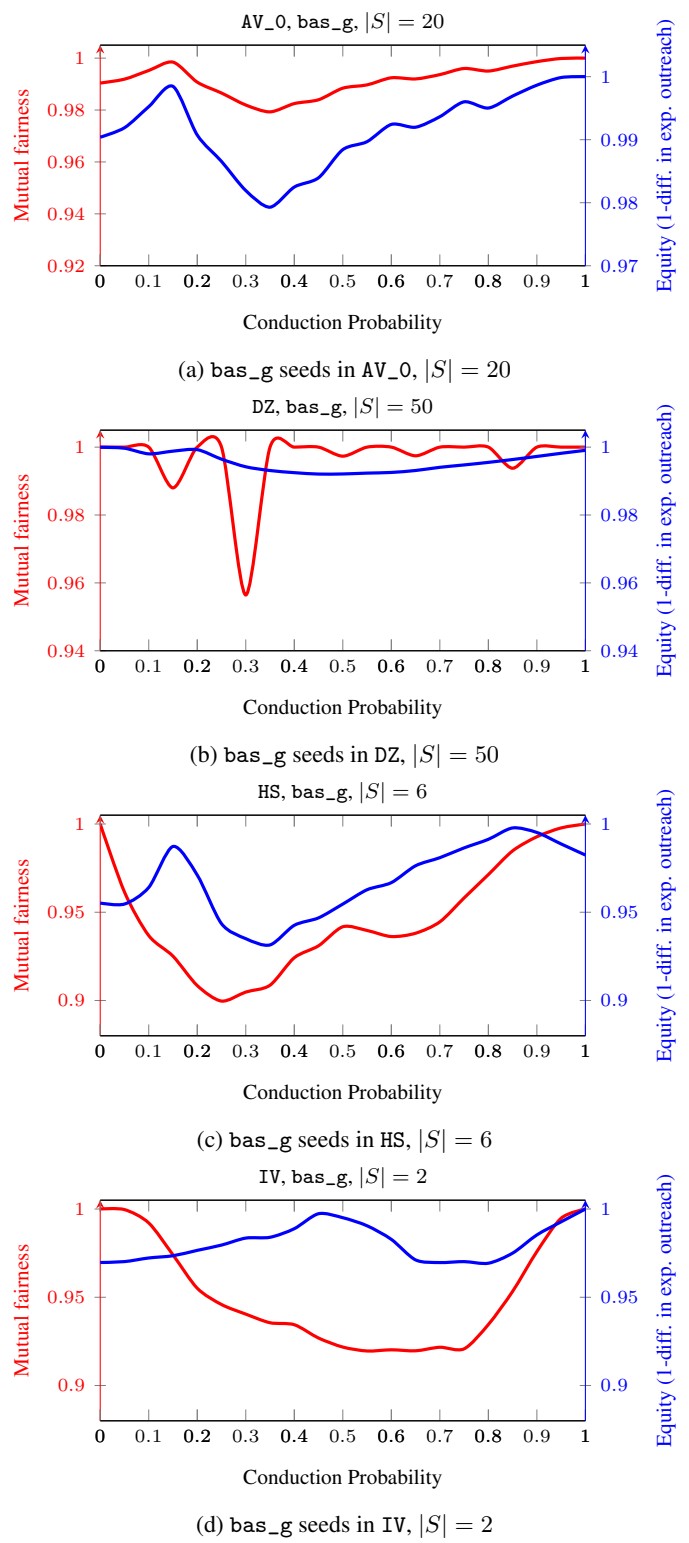

(a) `bas_g` seeds in `AV_O`, $|S| = 20$

(b) `bas_g` seeds in `DZ`, $|S| = 50$

(c) `bas_g` seeds in `HS`, $|S| = 6$

(d) `bas_g` seeds in `IV`, $|S| = 2$

Figure 13: Part 2: Different definitions of fairness VS conduction probability on an outreach distribution created by the `bas_g` heuristic.

## D.3 Fairness-Efficiency performance of seedset selection algorithms

We report more experiments in Fig. 14.

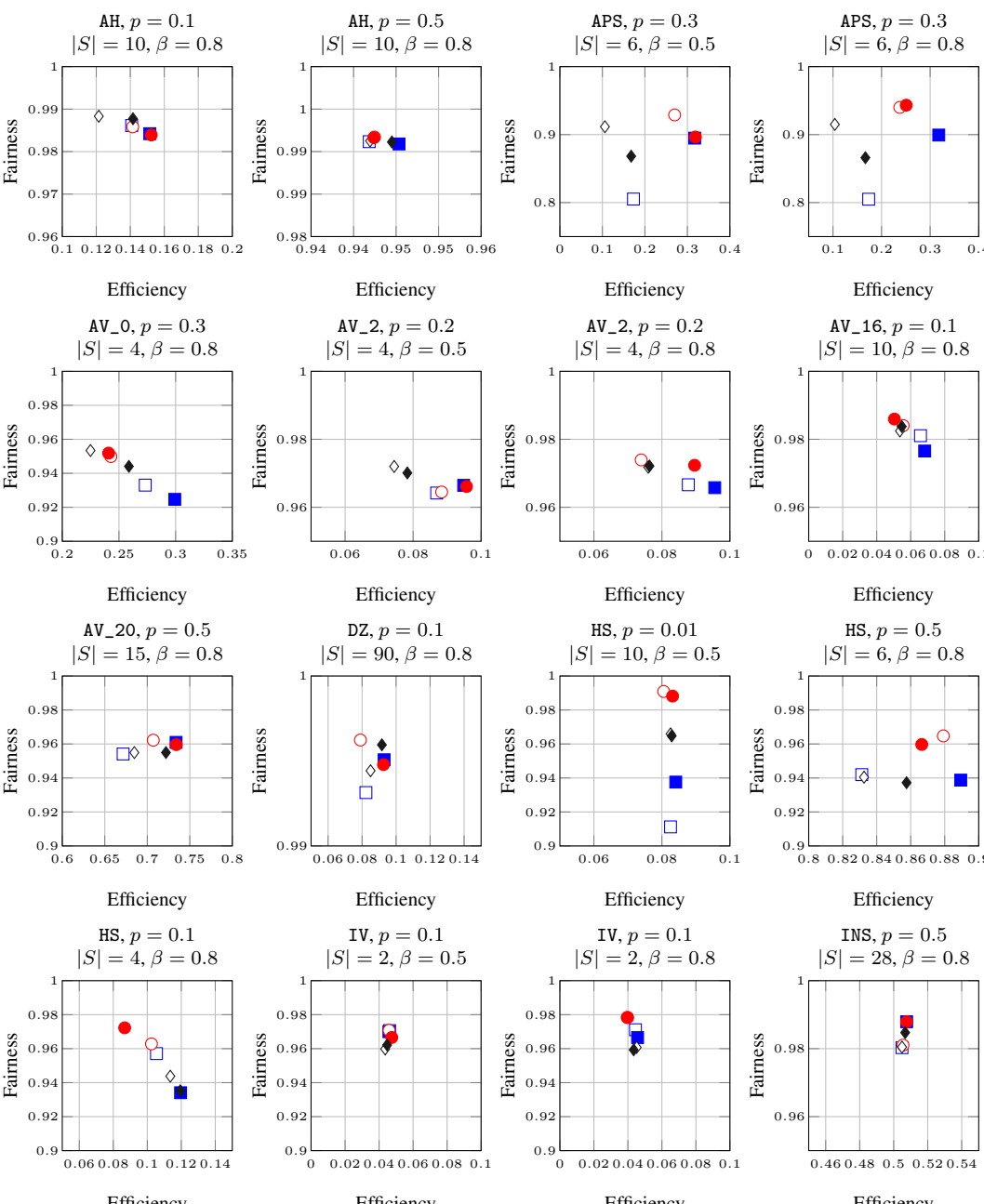

Figure 14: S3D trade-off and improvement against other label-aware and label-blind algorithms. Filled markers refer to greedy-based algorithms: ■ = bas_g, ● = S3D_g, and ♦ = hrt_g. Empty markers refer to degree-based algorithms: □ = bas_d, ○ = S3D_d, and ◇ = hrt_d.

# E  Details on the Algorithm

## E.1  Pseudocode

We provide more details on our algorithm, S3D, in two routines, Algorithm 2 and Algorithm 3.

---

**Algorithm 2** Seed Selection Stochastic Descent (S3D) Step: Pseudo Code

---

1: **function** SEEDSET_REACH(seedset,G,p,horizon)  ▷ nodes reached from seedset until horizon
2:     realizations ← 1000  ▷ for MCMC sampling, configurable
3:     reach ← []

4:     **while** realizations **do**
5:         reach ← reach+ INDEPENDENT_CASCADE(seedset, G, p, horizon)  ▷ collect nodes reached
6:         realizations ← realizations − 1

7:     **return** reach  ▷ repetition of nodes reached

8: **function** S3D_STEP(seedset, G, p, fair_to_efficacy) ▷ each step yields a new seedset
9:     exploit_to_explore ← 1.3  ▷ experimentally chosen, configurable
10:    non_acceptance_retention_prob ← 0.95  ▷ prob. of retaining set
11:    max_horizon ← GET_DIAM(G)
12:    horizon_factor ← max_horizon/4  ▷ limit runtime
13:    shallow_horizon ← max_horizon/horizon_factor

14:    num_seeds ← $len$(seedset)
15:    seedset ← DISTINCT(seedset)
16:    seedset ← FIT_TO_SIZE(seedset, num_seeds)  ▷ fit to size with random nodes
17:    reach ← SEEDSET_REACH(seedset, G, p, max_horizon)
18:    candidate_set ← [SAMPLE(reach, 1)]  ▷ get first in candidate seedset

19:    **while** num_seeds **do**
20:        last_seed ← candidate_set[−1]  ▷ get latest seed
21:        ▷ remove shallow reach of last seed from current reach
22:        reach ← reach− SEEDSET_REACH([last_seed], G, p, shallow_horizon)
23:        candidate_set ← candidate_set + [SAMPLE(reach, 1)]  ▷ extend new seedset
24:        num_seeds ← num_seeds − 1

25:    curr_score ← -BETA_FAIRNESS(seedset, fair_to_efficacy)
26:    candidate_score ← -BETA_FAIRNESS(candidate_set, fair_to_efficacy)

27:    ▷ Metropolis Sampling
28:    energy_change ← curr_score − candidate_score
29:    accept_prob ← CLIP(exp(exploit_to_explore ∗ energy_change), [0, 1])

30:    nonce_1 ← $U(0, 1)$
31:    **if** nonce_1 < accept_prob **then**
32:        **return** candidate_set  ▷ get a better seedset
33:    **else**
34:        nonce_2 ← $U(0, 1)$
35:        **if** nonce_2 < non_acceptance_retention_prob **then**
36:            **return** seedset  ▷ retain existing choice
37:        **else**
38:            random_set ← SAMPLE(G.nodes, num_seeds)
39:            **return** random_set  ▷ completely random selection rarely

---

### E.2 Estimating Runtime

We estimate the running time of Algorithm 2 and 3 combined. For the S3D_STEP, lines 9-13 are constant operations and comprise dataset properties. Lines 14-15 cost $O(|S|)$. FIT_TO_SIZE can cost up to $O(|S| \log |V|)$ for sampling new $|S|$ nodes. SEEDSET_REACH does repeated BFS, and so costs $O(R(|V| + |E|))$. Lines 19-24 cost as follows,

$$O((|S - 1|)(Rd_{\text{avg}}^{D_{\text{max}}} + R|V| + \log R|V|))$$

where $d_{\text{avg}}$ is the average degree of the graph, and $D_{\text{max}}$ is the largest diameter of the graph. The first term here upper bounds the max computation in BFS for $D_{\text{max}}$ horizon. Other terms follow

---
**Algorithm 3** S3D Iteration: Pseudo Code
---
```
 1: function S3D_ITERATE(seedset, G, p, fair_to_efficacy, num_iters)
 2:     least_score_seedset ← seedset
 3:     least_score ← -BETA_FAIRNESS(seedset, fair_to_efficacy)

 4:     while num_iters do
 5:         seedset ← S3D_STEP(seedset, G, p, fair_to_efficacy)
 6:         score ← −BETA_FAIRNESS(seedset, fair_to_efficacy)
 7:         if score < least_score then
 8:             least_seedset ← seedset
 9:         num_iters ← num_iters − 1

10:     return least_seedset
```
---

from the remaining operations in the `while` loop. Now, lines 25-26 first create an outreach from the corresponding seedsets, costing $O(R(|V| + |E|))$ each, and then analytically calculate $\beta$-fairness for all the $R$ final configurations, costing $O(R * 1)$ each. In the worst case, we might additionally execute lines 37-39 costing $O(|S| \log |V|)$. So, a single `S3D_STEP` costs

$$
\begin{aligned}
O(2|S| + |S| \log |V| &+ R(|V| + |E|) + (|S| - 1)(Rd_{\mathrm{avg}}^{D_{\max}} + R|V| + \log R|V|) \\
&+ 2(R(|V| + |E|) + R) + |S| \log |V|) \\
&= O(|S| \log |V| + R(|V| + |E|) \\
&+ R|S| + R|S||V| + |S| \log |V|) \\
&= O(|S| \log |V| + R|S||V|) \\
&= O(R|S||V|).
\end{aligned}
$$

Here, we used the assumption that $d_{\mathrm{avg}} = O(2E/V) = O(1)$ for a sparse graph ($E = O(V)$). Now this `S3D_STEP` is run $k$ times using `S3D_ITERATE` to find the best seedset in these $k$ runs. Moreover, we avoid any redundant calculations and memorize $\beta$-fairness for any seedset we discover. Hence, the total runtime is $O(kR|S||V|)$, as claimed.

### E.3 Motivation and Extension to Generic Combinatorial Optimization

The S3D approach to $\beta$-fairness optimization in this setting is independently motivated and in general can be extended to any combinatorial optimization problem where each choice of initial action at time $t = 0$, amongst exponentially many choices of actions, can lead a system to one of exponentially many states, for which we know the probability distribution of the system achieving one of these states and an associated, possibly non-convex, expected energy profile resulting from this stochastic state occupancy of the system at a later time. S3D then boils down to iteratively trying different initial actions that lead to small changes in the state occupancy distribution that align well with the ideal occupancy distribution, leading to a gradual reduction of the expected potential energy of the resulting system using Metropolis Sampling/Simulated Annealing.

In this study, the initial action at $t = 0$ is the initial seedset choice $S$, which leads to a distribution of states, called the outreach distribution on final configuration (Theorem 2.1), that the system, a Social Network here, can reach to. Each such distribution corresponds to a bounded expected "potential energy" (keeping the ideally mutually fair distribution as reference) defined on $\beta$-fairness– a mutually fair configuration is defined to be a "stable", less "energetic" system, and $S3D$ aims to achieve it via an optimal choice of $S$, $S^*$.

### E.4 Theoretical Guarantees of Convergence in S3D

The S3D algorithm is similar to non-convex optimization methods such as Simulated Annealing. Such algorithms do not have theoretical guarantees but have a long history of empirical success.

Let $f : \mathcal{P}(V) \to [0, 1]$ be the $\beta$-fairness set-evaluation function defined in the power set of the graph vertex set $V$. The function can then evaluate any seedset, $S \subseteq V$ for its $\beta$-fairness. Now for iterative optimization purposes, S3D defines a sampling process to define neighbors $S'$ of $S$, based on similar outreaches $V_{S'}$ and $V_S$. Then S3D essentially follows non-convex optimization of $f$ using *Simulated*

*Annealing* under *Metropolis Sampling* at a constant temperature. While Simulated Annealing does not have strict mathematical guarantees to find the global optimum in finite time, its empirical success is well understood in non-convex optimization.

While Simulated Annealing usually runs for finite iterations defined by an empirically tested temperature schedule, we ran Simulated Annealing under several constant temperatures to estimate the performance of S3D against baselines and concluded that a number of iterations $k \in [500, 1000]$ usually works well in practice. Hence, any decaying temperature schedule that translates total iterations in this range should work fine.

### E.5 Illustrative Example

Consider the information spreading over the graph in Fig. 15 as an independent cascade model with probability $p = 0.1$, with blue and red nodes belonging to two different groups. A greedy strategy would choose the seed set as $S_g = 3, 5$ (enlarged nodes) as shown in Fig. 15a, thus leading to the highly unfair outreach in Fig. 15b. In contrast, our algorithm S3D promotes the choice $S_{\text{S3D}} = 1, 4$ reflected in 15c, which gives the more fair outreach plotted in Fig. 15d, showing that it improves over greedy/sophisticated label-blind seed selection strategies.

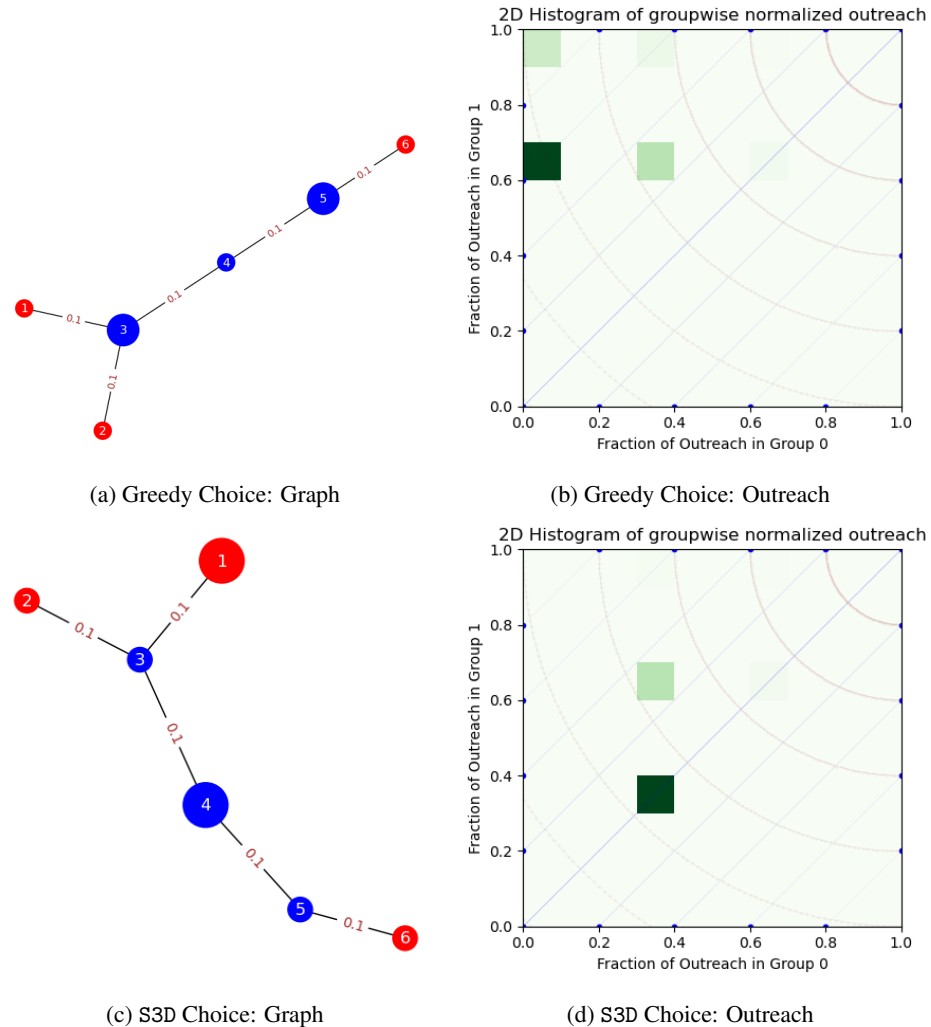

(a) Greedy Choice: Graph

(b) Greedy Choice: Outreach

(c) S3D Choice: Graph

(d) S3D Choice: Outreach

Figure 15: Toy example to show label-aware choice using S3D over a label-blind seedset selection process. The enlarged nodes are selected seeds. Since the graph is small, the outreach discretization bucket has been granularized for improved readability.

# F    Error Bars on Fairness and Efficiency Experiments

Referring to Fig. 7, we mention $2\sigma$ symmetrical error bars on $100$ repetitions for each of the experiment pipelines, as follows. Since each experiment itself runs on $R = 1,000$ realizations, we take $100R = 10^5$ samples of each random graph encoded social network dataset.

| - | Eff-Mean | Efficiency-Err-Bar ($\pm 2\sigma$) | Fair-Mean | Fairness-Err-Bar ($\pm 2\sigma$) |
|---|---|---|---|---|
| s3d_d | 0.24 | 0.0022 | 0.94 | 0.002 |
| hrt_d | 0.105 | 0.005 | 0.911 | 0.004 |
| bas_d | 0.173 | 0.0016 | 0.803 | 0.003 |
| s3d_g | 0.25 | 0.002 | 0.945 | 0.002 |
| hrt_g | 0.17 | 0.0058 | 0.868 | 0.006 |
| bas_g | 0.318 | 0.002 | 0.898 | 0.003 |

Table 2: APS.

| - | Eff-Mean | Efficiency-Err-Bar ($\pm 2\sigma$) | Fair-Mean | Fairness-Err-Bar ($\pm 2\sigma$) |
|---|---|---|---|---|
| s3d_d | 0.241 | 0.005 | 0.95 | 0.002 |
| hrt_d | 0.227 | 0.005 | 0.951 | 0.002 |
| bas_d | 0.277 | 0.004 | 0.935 | 0.002 |
| s3d_g | 0.241 | 0.005 | 0.951 | 0.002 |
| hrt_g | 0.258 | 0.005 | 0.945 | 0.003 |
| bas_g | 0.3 | 0.004 | 0.926 | 0.003 |

Table 3: AV_0.

| - | Eff-Mean | Efficiency-Err-Bar ($\pm 2\sigma$) | Fair-Mean | Fairness-Err-Bar ($\pm 2\sigma$) |
|---|---|---|---|---|
| s3d_d | 0.08 | 0.0004 | 0.99 | 0.0008 |
| hrt_d | 0.08 | 0.0004 | 0.967 | 0.0007 |
| bas_d | 0.08 | 0.0004 | 0.91 | 0.001 |
| s3d_g | 0.08 | 0.0004 | 0.988 | 0.0008 |
| hrt_g | 0.08 | 0.0004 | 0.965 | 0.0008 |
| bas_g | 0.08 | 0.0004 | 0.938 | 0.0009 |

Table 4: HS, $p = 0.01$.

| - | Eff-Mean | Efficiency-Err-Bar ($\pm 2\sigma$) | Fair-Mean | Fairness-Err-Bar ($\pm 2\sigma$) |
|---|---|---|---|---|
| s3d_d | 0.88 | 0.002 | 0.96 | 0.001 |
| hrt_d | 0.83 | 0.002 | 0.94 | 0.002 |
| bas_d | 0.83 | 0.002 | 0.94 | 0.002 |
| s3d_g | 0.87 | 0.002 | 0.96 | 0.002 |
| hrt_g | 0.86 | 0.002 | 0.935 | 0.002 |
| bas_g | 0.89 | 0.002 | 0.94 | 0.002 |

Table 5: HS, $p = 0.5$.

# G   Declaration of Computational Resources

All experiments were performed on a local PC on a single CPU core $3.5$ GHz. Except for datasets DZ, INS, all datasets were loaded and operated on a local PC with 32 GB of RAM. For the largest datasets (DZ, INS), we used remote compute clusters with $\sim 64$ GB memory and similar CPU capabilities. For the code development, we broadly used Python 3.10+, numpy, jupyter, and networkx [8]. Runtime for each non-S3D configured experiment on datasets except DZ, INS, was $10 - 15$ minutes. For DZ, INS, this was approximately $1 - 2$ hours. For S3D optimizations to be satisfactory, we ran each small dataset (except DZ, INS) for $1.5$ hours additionally. For massive datasets DZ, INS, the cluster took $\sim 4$ days for $k = 10$ steps. The total set of experiments made, including the failed, passed or submitted ones, roughly took the same order of resources separately.

