# OpenReview forum: "Fairness in Social Influence Maximization via Optimal Transport"
_NeurIPS.cc/2024/Conference — NeurIPS 2024 poster_

### Official Review · Reviewer_1BeY · 2024-07-02

**Soundness:** 2
**Presentation:** 3
**Contribution:** 3
**Rating:** 5
**Confidence:** 4

**Summary:**

This paper proposed a new metric for fairness in social influence maximization, namely mutual fairness. It is based on optimal transport theory. A parameter \beta is designed to achieve a balance between fairness and efficiency. When \beta = 0, it ignores mutual fairness. When \beta = 1, it enforces mutual fairness and ignores efficiency. A selection algorithm S3D is proposed based on the mutual fairness.

**Strengths:**

The new metric for fairness in influence maximization is refreshing and innovative.

Examples make the presentation easy to understand.

**Weaknesses:**

1. In Definition 2.1, V is the network. Late in section 3.2, V is the set of nodes.

2. Consider two influence strategies, A and B, for a population of two groups, I and II. Strategy A always informs everyone in Group I while ignoring everyone in Group II. Strategy B is the opposite. The optimal transport discrepancy between the two probability measures is \sqrt{2}. But they have the same mutual fairness score. This is a little counter-intuitive. Maybe try to modify it.

3. The generalizability from two groups to multi-groups should be demonstrated in detail.

**Questions:**

I still don't understand why the example scenario is unfair.

**Limitations:**

Limitations are discussed.

---

> ### Author Rebuttal · Authors · 2024-08-06
>
> # W1
>
> Thanks for pointing this out. We made the proposed correction in the manuscript.
>
> # W2
>
> Strategy A and Strategy B have the same mutual fairness score since they are equally (un)fair. Namely, they have the same distance from the diagonal. On the other hand, transporting Strategy A onto Strategy B has the highest cost as they are at a maximum (Euclidean) distance apart from each other. In a sense, the two strategies are ``symmetric'' in terms of unfairness. We would be glad to exemplify this further in the final version of the paper.
>
> # W3: The generalizability to multi-groups
>
> We agree with the reviewer that the multi-group case requires more detail. In future versions of the manuscript, we will include a dedicated appendix for the $m$-group case, which we summarize below.
>
> In the case of $m$ groups, the outreach distribution is a probability distribution $\gamma$ on the hypercube $[0,1]^m$; i.e., $\gamma$ now lives in $\mathcal P([0,1]^m)$.
> The reference distribution can again be taken to be the ``ideal'' distribution $\gamma^\ast=\delta_{(1,\ldots,1)}$ which encodes the case in which all members of all groups receive the information. As for the transportation cost, we can define it to be to distance between any given point $x=(x_1,\ldots,x_m)$ in the hypercube and the diagonal line. We can call this distance $d(x)$. An expression of $d$ can be found via simple geometric arguments. With this the fairness metric becomes
> \begin{equation*}
>     \textsf{Fairness}(\gamma)
>     =
>     1-
>     a
>     \mathbb{E}_{x\sim\gamma}[d(x)],
> \end{equation*}
> where the constant $a>0$ is again chosen so that $\textsf{Fairness}(\gamma)$ is between 0 and 1. Then, $\beta$-$\textsf{Fairness}(\gamma)$ can be defined analogously.
>
> Finally, since we once again provide a closed-form expression, there is no need to numerically solve optimal transport problems, as in the 2D case presented in the paper. Also, note that we do not resort to the multimarginal optimal transport problem, which would indeed cause exponential complexity in the dimensionality.
>
> # Q1: I still don't understand why the example scenario is unfair.
>
> The proposed example refers to the final configuration
> \begin{equation*}
>     \gamma_b = 0.25 \cdot \delta_{(0,0)} + 0.25 \cdot \delta_{(1,1)} + 0.25 \cdot \delta_{(0,1)} +0.25 \cdot \delta_{(1,0)}
> \end{equation*}
> where $\delta_{(i,j)}$ indicates the delta distribution centered at $(i,j)$, $i,j\in[0,1]$.
> The distribution $\gamma_b$ encompasses the case in which in 25\% of the cases anyone in Group 1 receives the information and none in Group 2 receives it ($0.25 \cdot \delta_{(1,0)} $) and in 25\% of the cases anyone in Group 2 receives the information and none in Group 1 receives it ($0.25 \cdot \delta_{(0,1)} $). Therefore, in 50\% of the cases, everyone in one group receives the information, as none in the other group receive it ($0.25 \cdot \delta_{(1,0)} + 0.25 \cdot \delta_{(0,1)}$).
>
> This represents a highly unfair outcome: referring to the example proposed by the reviewer, this means that half of the time we are in the situation of Strategy A or Strategy B, namely half of the time, there is one group that has no access to the information spread. We hope we have been able to clarify the doubt.
>
> We thank the reviewer for the time spent reviewing our paper and for their constructive feedback.

---

> ### Author Response · Authors · 2024-08-13
> **Comment from Reviewer 1BeY disappeared**
>
> Dear Area Chair,
>
> we wanted to notice you about  this comment
>
> "I thank the authors for providing clarification and the extra explanation on the generalizability to mullti-groups. The rebuttal solved most of my concerns. I still don't quite agree with Q1, but it is not critical in the paper. Thus, I would like to raise the score to 5."
>
>  from reviewer 1BeY.  We are not sure if there is any technical problem on the platform but the comment is not visible to us anymore. However, we can still see the comment among our notifications.
>
> Thank you very much for your help.
>
> The Authors

---

### Official Review · Reviewer_rBPP · 2024-07-02

**Soundness:** 4
**Presentation:** 4
**Contribution:** 4
**Rating:** 8
**Confidence:** 5

**Summary:**

The paper addresses the challenge of ensuring fairness in social influence maximization, where the goal is to select seed nodes in a social network to spread information equitably among different communities. The authors identify the limitations of existing fairness metrics, which often fail to account for the stochastic nature of information diffusion. To address this, they propose a new fairness metric, termed "mutual fairness," based on optimal transport theory. The authors also develop a seed-selection algorithm that optimizes both outreach and mutual fairness. Empirical evaluations on several real-world datasets demonstrate the efficacy of the proposed approach.

**Strengths:**

1.Novel Fairness Metric: The introduction of the mutual fairness metric represents a significant contribution. This metric captures the variability in outreach and provides a more accurate assessment of fairness in stochastic diffusion processes.
2.Practical Relevance: The proposed metric and algorithm are highly practical, addressing real-world issues in social influence maximization. The approach is shown to be effective across various datasets, making it a valuable tool for practitioners.
3.Thorough Evaluation: The authors conduct extensive experiments on diverse real-world datasets, demonstrating that their approach not only enhances fairness but also maintains or even improves efficiency.

**Weaknesses:**

Potential Scalability Issues: While the proposed method performs well on the datasets tested, its scalability to very large networks or to scenarios with many groups is not fully explored. Further analysis on the computational complexity and scalability would be beneficial.

**Questions:**

How does the mutual fairness metric perform in extremely large networks, and what are the computational challenges associated with scaling the approach?

**Limitations:**

Network Topology: Expand the experiments to include a wider variety of network topologies, providing insights into how different structures impact the fairness and efficiency of the approach.

---

> ### Author Rebuttal · Authors · 2024-08-06
>
> # W1(/Q1): Potential Scalability Issues
>
> We thank the reviewer for the positive feedback and insightful comments. We first address both scalability aspects as follows:
>
> **Definition of fairness metric with $m$-groups**:
>
> We agree with the reviewer that the multi-group case requires more detail. In future versions of the paper, we will include an appendix for the $m$-group case, which we summarize below.
>
> In the case of $m$ groups, the outreach distribution is a probability distribution $\gamma$ on the hypercube $[0,1]^m$. The reference distribution can again be taken to be the ``ideal'' distribution $\gamma^\ast=\delta_{(1,\ldots,1)}$ which encodes the case in which all members of all groups receive the information. As for the transportation cost, we can define it to be to distance between any given point $x=(x_1,\ldots,x_m)$ in the hypercube and the diagonal line. We can call this distance $d(x)$. An expression of $d$ can be found via simple geometric arguments. With this the fairness metric becomes
> $$\textsf{Fairness}(\gamma)=1-a\mathbb{E}_{x\sim\gamma}[d(x)],$$
> where the constant $a>0$ is again chosen so that $\textsf{Fairness}(\gamma)$ is between 0 and 1. Then, $\beta$-$\textsf{Fairness}(\gamma)$ can be defined analogously. Finally, since we once again provide a closed-form expression, there is no need to numerically solve optimal transport problems, as in the 2D case presented in the paper. Also, note that we do not resort to the multimarginal optimal transport problem, which would indeed cause exponential complexity in the dimensionality.
>
>
> **Scaling with Large Graphs**
>
> Appendix D.2 details the computational complexity of the algorithm, assuming sparse graphs. Additionally, we tested on a range of datasets summarized in Appendix B::Tab. 1. When considering sparse datasets, we see a linear growth of the computational complexity with the number of graph nodes. While we still accommodate time-complexity growth with the growth of the seedset size $|S|$, the SIM problem in the literature usually keeps them small subject to its definition. $R=1000$ worked well across all datasets (see Appendix E::Tab. 2-5 for error-bars). Moreover, if the graphs aren't sparse (as we tested with dataset DZ), computation time grows additionally with $|E|$ as highlighted in App. D.2.
>
> In practice, for datasets like INS with ~500,000 nodes and edges, generating a new seedset candidate can take around ~5 hours on computational resources declared in App. F, and is the usual bottleneck in the entire algorithm.
>
> **Computational precision in approximating joint distribution**
>
> Approximating outreach distribution is also limited by the extent to which we discretize the probability space. Added to this, if the graph is dense enough to present numerically close joint outreach sample points, we are forced to bucket them into a single support point in the probability space, trivializing the optimization problem and making similar seedset selections look equally fair and efficient.
>
> Moreover, one can see exponential growth in handling the discrete support space with the number of groups $m$, if we increase the precision/discrete bucket size of the support space. For our experiments, discretizing $[0, 1]^m, m=2$ space with $100$ buckets for each group meant handling $100^2=10^4$ points in probability space.
> Nonetheless, since we provide a closed-form expression for the fairness metric, bypassing the computation of the optimal transport problem, its evaluation does not suffer from an increased discretization.
>
> We will include a detailed discussion on time complexity and scalability in the final version of the paper.
>
> # L1: Network Topology
>
> **Network topology:** We discuss the impact of network topology in Sec. 3.2, and we will expand on this point in the final version of the paper. In short, when the conduction probability is too small or too large, network topology does not play a major role. When the conduction probability is moderate, network topology starts playing a role, mainly through the number of cross-group edges (CE):
>
> CE% is small (~5%, APS): Such datasets encode group interaction information in the edges themselves, that is, an edge likely means nodes belong to the same group. In such cases, baseline greedy algorithms (bas\_g) already perform well as they rely only on the edge connectivity which is extremely reliable here. S3D does not significantly improve on their selection, both in efficiency and fairness.
>
> CE% is balanced (40-50%, HS, AH): These datasets reflect that groups interact well across each other and hence any seedset selection largely ends up getting a fair outreach. Since bas\_g already has proven near-optimal efficiency guarantees too, S3D performing significantly better than bas\_g is again unlikely.
>
> CE% is moderate (5-30%, AV (datasets ids $0, 2, 16, 20$), IV): These are the non-trivial cases not covered above. And here bas\_g isn't lucky enough to reliably leverage the existence of edges into group information. Hence, S3D usually outperforms the baseline in these cases, achieving similar efficiency scores and significantly improving fair outreach through its seedset selection.
>
> CE\% is high (>50%): This case was never observed where nodes interact more across groups than in their own groups. However, as long as the existence of edges doesn't reliably signal group information, we expect S3D to perform well based on a similar analysis.
>
> **Moderate outreach in dense graphs (\INS, DZ):**
> Graphs where $|E|$ substantially exceeds $|V|$, the outreach variance across sample sub-graphs is too low to be captured in the discretized space we experimented in (100x100 units in  $[0,1]^2$), even for moderate $p$. This leads to single-point concentrated joint-distribution plots across several seedsets S3D tries to evaluate, all of them leading to the same OT score/$\beta$-fairness.
>
> We thank the reviewer for the positive assessment of our paper and their constructive feedback.

---

> ### Comment · Reviewer_rBPP · 2024-08-07
> **Accept**
>
> The introduction of the mutual fairness metric represents a significant contribution. And, the author addressed my issue.

---

### Official Review · Reviewer_vnZX · 2024-07-09

**Soundness:** 3
**Presentation:** 3
**Contribution:** 3
**Rating:** 7
**Confidence:** 3

**Summary:**

This paper studies the problem of Fair Social Influence Maximization (SIM). Specifically, it introduces a new notion of fairness for SIM. The current literature on Fair SIM studies defines fairness in terms of expected values, e.g., a solution is fair if the expected ratio of influenced nodes from each demographic group is the same. The paper at hand begins by demonstration scenarios where the aforementioned definitions of fairness fail to provide truly fair outcomes. Namely, scenarios where the expected values are not enough to capture fairness (50% chance of all red nodes getting influenced with 0 blue nodes influenced, and 50% chance of all blue nodes getting influenced with 0 red nodes influenced). To address and mitigate such obviously problematic cases, the authors introduce a novel fairness definition based on optimal transport. Fix the optimally fair joint distribution $\gamma^*$, in which both groups always receive the same ratio of influenced nodes. Then a joint distribution $\gamma$ is considered fair is it minimizes the transport distance to $\gamma^*$. The cost function used in the transport distance is the natural choice of a Euclidean distance. The authors call this notion of fairness mutual fairness.

After introducing mutual fairness, the authors proceed with an extensive experimental evaluation that demonstrates that algorithms which satisfy notions of expected fairness are not necessarily mutually fair. Moving forward, they show how the trade-off between fairness and efficiency (how many nodes are reached in total) can be naturally incorporated in their definition of mutual fairness; this gives rise to a new definition called $\beta$-fairness where $\beta$ is a parameter controlling the aforementioned trade-off. Finally, the authors present an algorithm that is specifically designed to optimize mutual fairness. Their experimental evaluation shows that this algorithm dominates existing baselines.

**Strengths:**

1) Very interesting novel concept of fairness for SIM.
2) Solid results with extensive experimental evaluation.
3) Excellent presentation.

**Weaknesses:**

1) Lack of theoretical guarantees for the S3D algorithm.

**Questions:**

Shouldn't $\gamma$ be $\pi$ in line 184, in the definition of mutual fairness?

---

> ### Author Rebuttal · Authors · 2024-08-06
>
> # W1: Lack of theoretical guarantees for the S3D algorithm
>
> We thank the reviewer for the positive feedback. To first address the point regarding the lack of theoretical guarantees: The S3D algorithm is similar to non-convex optimization methods such as Simulated Annealing. Such algorithms do not have theoretical guarantees but do have a long history of empirical success. We think that obtaining theoretical guarantees would be an excellent avenue for future work.
>
> For further details on S3D, let $f: \mathcal{P}(V) \rightarrow [0, 1]$ be the $\beta$-fairness set-evaluation function defined in the power-set of the graph vertex set $V$. The function can then evaluate any seedset, $S \subseteq V$ for its $\beta$-fairness. Now for iterative optimization purposes, S3D Algorithm 1::3-8 defines a sampling process to define neighbors $\hat{S}$ of $S$ through its outreach $V_S$. Then S3D essentially follows non-convex optimization of $f$ using Simulated Annealing under Metropolis Sampling at a constant temperature. While Simulated Annealing does not have strict mathematical guarantees to find the global optimum in finite time, its empirical success is well understood in non-convex optimization.
>
> While Simulated Annealing usually runs for finite iterations defined by an empirically tested temperature schedule, we ran Simulated Annealing under several constant temperatures to estimate the performance of S3D against baselines and concluded that a number of iterations $k \in [500, 1000]$  usually works well in practice. Hence any decaying temperature schedule that translates total iterations in this range should work fine.
>
> # Q1
>
> The optimal transport discrepancy $W(\gamma,\gamma^*)$, appearing in line 184, is computed between two probability distributions: the probability measure $\gamma$ and the desired probability measure $\gamma^*$. The symbol $\pi$, instead, refers to the transportation plan and it is the optimization variable associated with the optimal transport problem. In line 184, the explicit expression of the solution of the optimization problem is already given, for this reason, $\pi$ does not appear. The subscript $(x_1,x_2)\sim \gamma$, instead, indicates that the samples $x_1,x_2$ are drawn from the probability distribution $\gamma$. We will further clarify these aspects in the text.
>
> We thank the reviewer for the positive assessment of our paper and their constructive feedback.

---

### Official Review · Reviewer_wcrW · 2024-08-01

**Soundness:** 3
**Presentation:** 2
**Contribution:** 3
**Rating:** 5
**Confidence:** 5

**Summary:**

This paper studies the problem of fairness in IM (Influence Maximization). They present a novel notion of fairness, namely, mutual fairness, which considers outreach distribution in different groups. Compared with previous notions, the proposed one could ensure a higher probability of fairness among groups by evaluating fairness via optimal transport. Based on mutual fairness, they propose the S3D (Stochastic Seedset Selection Descent) algorithm that shows better performance in experiments.

**Strengths:**

**S1**: Presenting a novel notion of fairness in IM which seems really appealing to me.

**S2**: Evaluating the level of fairness via optimal transport.

**S3**: Presenting the S3D algorithm.

**Weaknesses:**

**W1**: In this paper, the authors only consider $m=2$ groups and claim the framework is easily generalizable to more groups. I doubt about this in two aspects. On the one hand, the case would become much more complex (growing exponentially) when considering multiple variable probability distributions. On the other hand, the distance between two distributions can be easily calculated when there are only two groups. I wonder how to calculate such distance when the distribution contains $m$ groups.

**W2**: The example in motivation and the motivating example seem to be rather extreme cases. I can understand such examples are only constructed to show the significance of mutual fairness, but these cases can hardly occur in real-world scenarios.

**W3**: Since this paper evaluates fairness based on utility distribution, how to depict the ground-truth utility distribution with high approximation probability is of great importance. The authors should discuss this point. Besides, $R$ times of Monte Carlo simulation with $R=1000$ is far from enough in the field of IM. Please check related references.

**W4**: The authors claim that "the equity metric fails to adequately capture changes in fairness" based on the results in Figure 4. However, the y-axis in Figure 4 only ranges from 0.9 to 1. Therefore, both metrics are already high enough to reflect their deep difference. Also, in Appendix C.2, the two metrics are even more similar. The benefit of using mutual fairness is not so obvious.

**W5**: In Figure 5(d), the outreach of the proposed method is significantly higher than the Greedy. Does it really happen? Note that the Greedy has a theoretical guarantee of $(1-1/e-\varepsilon$)-approximation.

**Questions:**

In addition to the weaknesses I mentioned above, I also have some minor concerns.

**Q1**: Please explain $\delta_{(i,j)}$ in the paper.

**Q2**: Eq. (2) has a mistake. I think the later part should be $\sqrt(2)/2 \cdot |(x_2-x_1)-(y_2-y_1)|$ rather than $|(x_2-x_1)-(y_1-y_2)|$ since it is Euclidean distance between $z$ and $(x_1,x_2)$. The sample problem also happens in Eq. (3).

**Q3**: The authors should state clearly that which notion is $hrt_g$ based on, equity or equality?

**Q4**: In Figure 4, I believe that the y-axis should be $1-diff.$ in exp. outreach.

**Q5**: When $\beta=0$, does $\beta$-fairness degenerate to the classic IM problem?

**Q6**: Please explain the "fixed horizon" in Algorithm 1.

**Q7**: I suggest the authors use $S$ and $S_o$ for candidates and the initial seed set for a clearer presentation. The variant font of $S$ could easily lead to readers' confusion.

**Q8**: I am confused by the weird phenomenon where results in Figure 5(b-d) appear as discrete rectangles. Any explanation?

**Limitations:**

The authors have addressed their limitations. However, I am wondering whether the method can be **easily** generalized to more groups, as I mentioned in Weakness.

---

> ### Author Rebuttal · Authors · 2024-08-07
>
> # W1
> We agree that the multi-group case requires more detail. We will include a dedicated appendix. With $m$ groups, the outreach distribution is a distribution $\gamma$ on $[0,1]^m$. The reference distribution is again the ``ideal'' distribution $\gamma^\ast=\delta_{(1,\ldots,1)}$ which encodes the case in which all members of all groups receive the information. As for the transportation cost, we can define it to be to distance between any given point $x=(x_1,\ldots,x_m)$ and the diagonal line. We can call this distance $d(x)$. An expression of $d$ can be found via simple geometric arguments. Then, $$\textsf{Fairness}(\gamma)=1-a E_{x\sim\gamma}[d(x)]$$ where $a>0$ is so that the metric is between 0 and 1. Then, $\beta$-fairness can be defined analogously. Since we have a closed-form expression, there is no need to numerically solve OT problems, as in the paper. Also, we do not resort to multimarginal OT, which would cause exponential complexity in $m$.
> # W2
> The example we draw inspiration from to design our metric, however, is an actual occurrence in our experiments. For the APS dataset with greedy fair heuristic, which exploits the notion of equity (Fig 3c in the paper, Fig a in the PDF), the samples are distributed in a way such that a consistent percentage (~30%) of the members in group 1 receive the information and way fewer members of group 2 receive it, and vice-versa. We show in Fig 5a that our algorithm can counteract unfair occurrences by moving the distribution closer to the diagonal (Fig b in the PDF). We will put more emphasis on this in the final version.
> # W3
> We agree that the $R$ leading to a statistically strong result depends on the dataset: the varied geometry of various subgraphs that the sampling of the edges produce, the exact complexity of the joint-outreach distribution (ground-utility distribution in this context), and the uncertainty of the quantities we sample/approximate throughout our experiments. App. E: Tab. 2-5 highlight the level of uncertainty, being < 3% in $2\sigma$ error-bar. To generate these bars using the $R$ values, we run each configuration 100 times, leading to instances using $100R = 10^5$ independent subgraph samples, sufficiently more than/of the same order in the relevant literature. We will highlight this in the revision.
> # W4
> We show earlier in the paper that our metric is more informative than the equity metric, both mathematically (Sec 3.1) and with a vast amount of experiments (Sec 3.2). Fig 4 confirms these findings by noticing that the two metrics show significantly different trends across different propagation probabilities, rather than their absolute closeness. E.g., in Fig 4, the higher conduction prob. makes the same setup "relatively" unfair when seen from our metric's lens, totally opposite of what expected outreach highlights, establishing a non-trivial and fundamental difference in them.
> # W5
> The reviewer is correct. There was a mislabeling in Fig 5b-d. Fig 5b is a comparison between the greedy heuristics versus S3D with greedy baseline and Fig 5d is a comparison between the degree centrality heuristics versus S3D with degree centrality baseline. The behavior in Fig 5d is now in sync with the corresponding metric dots in Fig 6's last subfigure. This also agrees with our understanding that the greedy strategy already has a theoretic efficiency lower-bound guarantee, unlikely to be surpassed (see Fig. 6 last subfigure). What S3D achieves is a tradeoff for ~1% efficiency to get a >2% gain in fairness against greedy heuristics. Whereas against degree centrality heuristics, there is a substantial gain in both fairness and efficiency. We have corrected the figures in the PDF.
> # Q1
> It is the delta distribution at $(i,j)\in [0,1]^2$. We will mention this in the final version.
> # Q2
> Thanks for pointing out this inaccuracy. Indeed, the transportation cost is $$\Vert z(x_1,x_2,y_1,y_2)-x\Vert=\frac{\sqrt{2}}{2}|(x_2-x_1)-(y_2-y_1)|.$$ The factor $\sqrt{2}/2$ is then dropped for normalization purposes, to ensure that the metric is between 0 and 1. In our applications, the reference distribution lies on the diagonal ($y_1=y_2$) which (after normalization) yields the same expression $|x_1-x_2|$. Thus, this inaccuracy does not affect our experiments. We detailed the computations and fixed Eq (2)-(3) in the revision.
> # Q3
> The alg. hrtg is based on equity. We will mention it in the paper.
> # Q4
> The reviewer is correct.
> # Q5
> Correct. We will mention this in the revision.
> # Q6
> When we consider information propagation from $S$ to generate the list $V_S$ of nodes reachable from $S$ (line 3), we use this $V_S$ to generate a candidate seedset $\hat{S}$ using iteration in lines 5-8, adding a new seed per step. This seed in each step, $v \sim V_S$, incrementally reduces the list $V_S$ by removing nodes from it that are reachable from $v$. Ideally, the reachability of $v$ alone is again defined by the nodes reachable via Independent Cascade (IC) starting from this seed. So, one might end up running IC $|S|$ many times, for all $v\in\hat{S}$. To approximately solve this problem, we create $V_{\hat{S}}$ (line 7) only considering nearest neighbors up to a constant depth from $v$ (fixed iterations of BFS from $v$). This constant is a fraction of the largest graph diameter.
> # Q7
> We appreciate the suggestion and will implement it.
> # Q8
> This behavior can also be seen well in the example in Fig. 14, where the reasoning is instantly perceptible. For visualization/computation reasons, $[0,1]^2$ is discretized into 100x100 (real-world datasets) or 10x10 (Fig 14) blocks. Meanwhile, the possible outreach fraction realizations (rational numbers), depending on the graph geometry and the number of graph nodes and selected seeds, are bucketed into the closest buckets. So, the visualization may appear as a pattern of "rectangles" which represent the buckets of the discretized space. These are more prominent with smaller datasets.
> # Limitations
> Please refer to W1.

---

### Author Rebuttal · Authors · 2024-08-07

Dear Chairs,

Dear Reviewers,


We thank you  for the thoughtful feedback on our manuscript.
In the original submission, all four Reviewers found our results of interest to the wide readership of NeurIPS.
In particular, all reviewers agree on the fact that the proposed fairness metric is innovative and of practical relevance.
The main reservation of the Reviewers concerns the extension of the new metric to multiple groups, which was originally limited to two groups and hampers its wide application to real-world complex systems.
Other suggestions to further improve the manuscript include clarification about the realism of the motivating example, the influence of the network topology, and practical scalability issues.

We have now revised the manuscript to address these and all other comments from the Reviewers.


In particular,

1) we have explicitly mentioned now how the method can be extended to multiple groups and, in particular, explained why our fairness metric does not suffer from exponential computational complexity in the number of groups, which one might expect from the multi-marginal formulation of optimal transport;

2) we have extensively listed how the cross-edge percentage affects fairness and efficiency compared to the baseline algorithms, by splitting the analysis into small, balanced, moderate, and high cross-edge percentages;

3) we have shown  how our motivating example indeed is a practical occurrence in the experiments;

4) we have elucidated what are the scalability issues of our algorithm when it comes to very large networks.


These revisions significantly improved the presentation of our fairness metric and seed selection strategy by making it clear how our method extends to multiple groups and how the strategy could be, in principle, implemented in practice.

Yours Sincerely,

The Authors

---

### Decision · Program_Chairs · 2024-09-25

**Decision:**

Accept (poster)

**Comment:**

The paper studies fair influence maximization problem for influence propagation in social networks. It proposes a new metric, mutual fairness, to address the limitation of previous fairness matric under stochastic outcomes. The reviewers acknowledge the novelty of the proposed metric, as well as the technical contribution of applying the optimal transport theory to the task. The issues raised by the reviewers are mostly addressed by the authors during the rebuttal stage. I thus recommend acceptance to the paper.